# Understanding the biological processes of kidney carcinogenesis: an integrative multi-omics approach

Ricardo Cortez Cardoso Penha [ID][1], Alexandra Sexton Oates[1], Sergey Senkin [ID][1], Hanla A Park [ID][1], Joshua Atkins[2], Ivana Holcatova[3], Anna Hornakova [ID][4], Slavisa Savic [ID][5], Simona Ognjanovic [ID][6], Beata Świątkowska [ID][7], Jolanta Lissowska [ID][8], David Zaridze[9], Anush Mukeria[9], Vladimir Janout[10], Amelie Chabrier[1], Vincent Cahais[11], Cyrille Cuenin[11], Ghislaine Scelo[12], Matthieu Foll[1], Zdenko Herceg[11], Paul Brennan[1], Karl Smith-Byrne[2], Nicolas Alcala[1] & James D Mckay [ID][1 ✉]

## Abstract

**Biological mechanisms related to cancer development can leave distinct molecular fingerprints in tumours. By leveraging multi-omics and epidemiological information, we can unveil relationships between carcinogenesis processes that would otherwise remain hidden. Our integrative analysis of DNA methylome, transcriptome, and somatic mutation profiles of kidney tumours linked ageing, epithelial–mesenchymal transition (EMT), and xenobiotic metabolism to kidney carcinogenesis. Ageing process was represented by associations with cellular mitotic clocks such as epiTOC2, SBS1, telomere length, and *PBRM1* and *SETD2* mutations, which ticked faster as tumours progressed. We identified a relationship between *BAP1* driver mutations and the epigenetic upregulation of EMT genes (*IL20RB* and *WT1*), correlating with increased tumour immune infiltration, advanced stage, and poorer patient survival. We also observed an interaction between epigenetic silencing of the xenobiotic metabolism gene *GSTP1* and tobacco use, suggesting a link to genotoxic effects and impaired xenobiotic metabolism. Our pan-cancer analysis showed these relationships in other tumour types. Our study enhances the understanding of kidney carcinogenesis and its relation to risk factors and progression, with implications for other tumour types.**

**Keywords** Integrative Multi-omics Analysis; Kidney Cancer; Genomic Epidemiology; Cancer Biology; Tumour Microenvironment
**Subject Categories** Cancer; Chromatin, Transcription & Genomics; Genetics, Gene Therapy & Genetic Disease

## Introduction

Renal cell carcinoma (RCC) is the 16th most common cancer type worldwide, accounting for ~2% of all cancer patient deaths in 2020 (Sung et al, 2021). Clear cell RCC (ccRCC) is the most frequent histological subtype, accounting for ~75% of kidney cancer cases (Hsieh et al, 2017). The incidence rates of ccRCC are higher in high-income countries, particularly in central and northern Europe (Hsieh et al, 2017), with an increasing trend in global incidence (Huang et al, 2022). Risk factors associated with ccRCC include age, sex, obesity, hypertension, and tobacco smoking, although they collectively explain less than 50% of the newly diagnosed cases (Hsieh et al, 2017).

A better understanding of the underlying biological mechanisms associated with ccRCC tumours may provide new insights into disease aetiology and how the tumour progresses. Endogenous and exogenous exposures leave distinct molecular marks through the course of a lifetime, detectable at DNA and RNA levels. Recurrent DNA mutation patterns, or DNA mutational signatures, have been linked to endogenous (e.g., cellular ageing, *APOBEC* activity) and exogenous exposures (e.g., tobacco smoke, UV light, aristolochic acid) (Alexandrov et al, 2015; Alexandrov et al, 2020; Scelo et al, 2014; Senkin et al, 2024). Interestingly, our group recently demonstrated that certain ccRCC risk factors were not associated with DNA mutational signatures (Senkin et al, 2024), suggesting non-mutagenic pathways linked to risk factors in ccRCC. The DNA methylome is similarly impacted by exogenous and endogenous exposures (Herceg et al, 2018), with tobacco smoking, ageing, and somatic driver mutations provoking aberrant DNA methylation patterns (Chamberlain et al, 2022; Guida et al, 2015; Halaburkova et al, 2020; Horvath, 2013; Motzer et al, 2020; Ricketts et al, 2018; Şenbabaoğlu et al, 2016; TCGA, 2013). Transcriptome data have also been used to generate molecular signatures that capture a variety of biological processes, such as those linked to biological

[1]Genomic Epidemiology branch, International Agency for Research on Cancer/World Health Organization (IARC/WHO), Lyon 69366, France. [2]Cancer Epidemiology Unit, University of Oxford, Oxford, Oxford OX3 7LF, UK. [3]Institute of Public Health & Preventive Medicine, Charles University, Prague 15000, Czechia. [4]Institute of Hygiene and Epidemiology, Charles University, Prague 12800, Czechia. [5]Department of Urology, Kliničko-Bolnički Centar Dr Dragiša Mišović, Belgrade, Serbia. [6]International Organization for Cancer Prevention and Research, Belgrade 11070, Serbia. [7]Department of Environmental Epidemiology, Nofer Institute of Occupational Medicine, Łódź 90-950, Poland. [8]Maria Sklodowska-Curie National Research Institute of Oncology, Warszawa 00-001, Poland. [9]N.N. Blokhin Cancer Research Center, Moscow 115478, Russia. [10]Faculty of Health Sciences, Palacký University Olomouc, 77900 Olomouc, Czechia. [11]Epigenomics and Mechanisms Branch, International Agency for Research on Cancer/World Health Organization (IARC/WHO), Lyon 69366, France. [12]The Observational & Pragmatic Research Institute, Midview City 573969, Singapore. ✉E-mail: mckayj@iarc.who.int

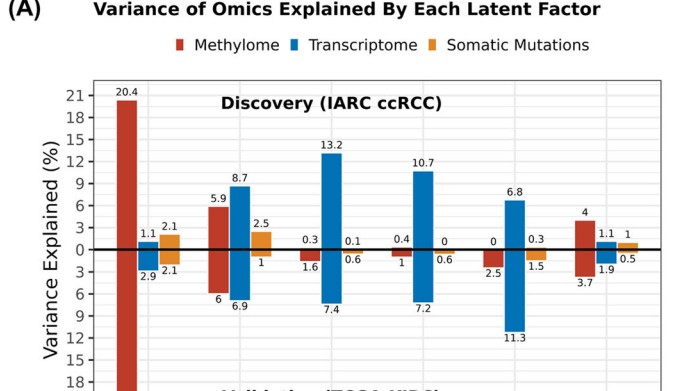

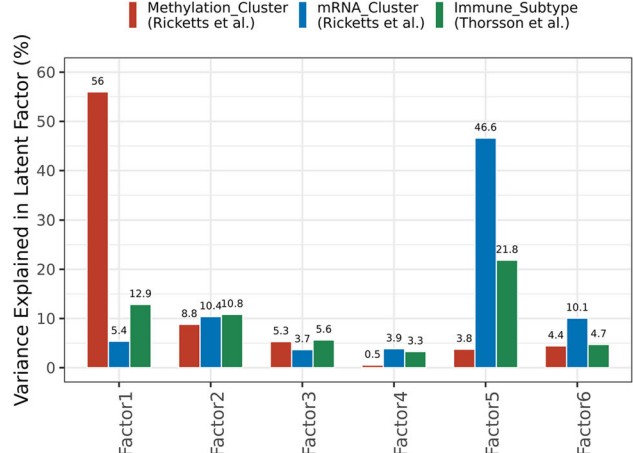

**Figure 1.  Overview of latent factors 1–6.**

(A) Percentage of variance in each omic layer across tumour samples from the discovery set (upper bars; all molecular variables included in the Multi-Omics Factor Analysis; IARC ccRCC: $N = 151$ tumours) and validation dataset (bottom bars; latent factor signatures; TCGA-KIRC: $N = 324$ tumours) explained by latent factor. Methylome (DNA methylation) in red, transcriptome (microarray) in blue, and somatic mutations (cancer driver mutations and DNA mutational signatures derived from whole-genome sequencing data for the discovery set and whole-exome sequencing data for the validation set) in orange. (B) To estimate the percentage of variance explained in each latent factor (factors 1–6) explained by the single-omic clusters derived from previous TCGA studies, we applied linear regression models where each latent factor was the outcome, and each omic cluster derived from previous TCGA studies was the predictor. This included three DNA methylation clusters and four mRNA clusters (Ricketts et al, 2018), as well as six expression-based immune subtypes (Thorsson et al, 2018). The models provided adjusted $R^2$ values, representing the variance explained by each omic cluster in the respective latent factor.

ageing (Peters et al, 2015), tumour microenvironment (TME) (Li et al, 2022; Thorsson et al, 2018), and cellular proliferation (Wolf et al, 2014).

Multi-omics kidney cancer studies using DNA methylation, gene expression, and somatic mutation profiles have been useful for describing molecular groups related to patient classification in relation to histological subtypes and outcome (Motzer et al, 2020; Ricketts et al, 2018; TCGA, 2013; Thorsson et al, 2018) while providing less resolution on disease aetiology. These previous studies generally focused on analysing each omic layer separately and integrating it afterwards. Integrative approaches that consider the covariance across different omics data were shown to improve the representation of the molecular processes present in somatic tissues compared with a single-omic approach (Argelaguet et al, 2018; Wolf et al, 2014). In the current study, we undertook integration of somatic DNA mutational signatures, cancer driver mutations, DNA methylome, and transcriptome profiles to describe sources of inter-patient variation from well-characterised ccRCC cohorts. We then triangulated sources of inter-patient variation with molecular and epidemiologic observations to explore the underlying biological processes related to the disease's aetiology and progression.

## Results

### Description of the molecular components in ccRCC tumours

The current work employed a two-stage study design (Fig. EV1) with discovery and validation ccRCC cohorts (Table EV1). An

initial discovery phase identified the sources of molecular variance or latent factors (LF) across tumours of ccRCC patients using the unsupervised Multi-Omics Factor Analysis (MOFA) to integrate DNA methylome, transcriptome, and somatic mutation profile data from whole-genome sequencing (WGS) data (Appendix Fig. S1). We then internally trained and tested LASSO regression models to select the most informative features of each LF in the ccRCC discovery cohort and build molecular signatures that represented the LF (Table EV2). These ccRCC LF signatures were used to evaluate these sources of variation in independent cohorts of ccRCC patients in a validation phase, as well as in other cancer types. Associations between LF and molecular annotations and epidemiological observations were first tested in the discovery series and then, wherever the variable was available, validated in the independent cohorts (Table EV1).

Collectively, 31%, 41%, and 6% of the variance in DNA methylome, transcriptome, and somatic mutation profile data, respectively, were explained by the first six LF estimated by MOFA in the discovery set. The proportion of variance explained in each omic layer remained similar when the LF signatures were inferred into the validation set (Fig. 1A). When compared to the results of PCA analysis derived from either DNA methylation or transcriptome data separately in the discovery cohort, only LF1 was resolved by a single-omic layer analysis (LF1 *vs.* DNAm_PC1, $R^2 = 0.96$), while L2-LF6 were partially resolved (Appendix Fig. S2). We estimated the percentage of variance in each LF explained by the single-omic clusters derived from previous TCGA studies using adjusted $R^2$ from linear regression models. This included three DNA methylation clusters and four mRNA clusters (Ricketts et al, 2018), as well as six expression-based immune subtypes (Thorsson et al, 2018). These single-omic clusters were partially correlated

with the LF, with 56% of the variance in LF1 being explained by the DNA methylation clusters, while 46% of the LF5 variance was explained by the mRNA clusters (Fig. 1B).

We then evaluated the relationship between LF and epidemiological data (Table 1; Fig. EV2). In the discovery phase, LFs were associated with chronological age (LF1: $P = 0.045$), sex (LF1: $P = 0.006$, LF6: $P = 0.010$), tobacco smoking (LF6: $P = 9 \times 10^{-6}$), alcohol consumption (LF3: $P = 0.020$), tumour stage (LF1: $P = 1 \times 10^{-5}$, LF2: $P = 3 \times 10^{-4}$, LF5: $P = 5 \times 10^{-4}$), and patient survival (LF1: $P = 0.022$, LF2: $P = 3 \times 10^{-4}$, LF5: $P = 0.002$). In the validation phase, the associations between chronological age (LF1: $P = 2 \times 10^{-5}$), sex (LF2: $P = 2 \times 10^{-5}$, LF6: $P = 9 \times 10^{-7}$), tumour stage (LF1: $P = 5 \times 10^{-18}$, LF2: $P = 2 \times 10^{-6}$, LF5: $P = 9 \times 10^{-7}$), and patient survival (LF1: $P = 4 \times 10^{-7}$, LF2: $P = 0.001$, LF5: $P = 0.003$) were replicated. The modest association with alcohol consumption was not replicated (LF3: $P = 0.94$, IARC validation set). The association between LF6 and self-reported tobacco smoking information was in a consistent direction but not statistically significant (LF6: $P = 0.16$), although analysis of this variable was hampered as it was available for only ~19% of patients in the validation set.

We also investigated the relationship between molecular features and LF in ccRCC tumours. The CpG sites associated with LF tended to be annotated to functional regions of the genome (CpG islands, shores, and shelves within regulatory and coding regions) in both discovery and validation sets (Figs. 2A and EV3A). There was little evidence for associations between LF and global DNA methylation changes, estimated by the mean DNA methylation levels of *Alu* and *LINE1* transposable elements (Table EV3). Pathway analysis showed that the gene expression levels with the highest loadings in each LF were enriched for cancer-related pathways, such as those involved on immune system, metabolism, cell cycle, cell plasticity and signalling, chromatin remodelling, and tissue development (Figs. 2B and EV3B; Table EV4). To explore which TME components correlate with LF, we imposed gene expression TME signatures derived from single-cell RNA sequencing (scRNA) data of ccRCC tumours (Li et al, 2022) into bulk ccRCC tumour transcriptome data used here (Appendix Fig. S3). LF2–6 were reproducibly associated with TME signatures related to kidney epithelial, immune cell infiltrates, inflammation, epithelial–mesenchymal transition process (EMT), and cell proliferation processes (Figs. 2B and EV3B). As a sensitivity test, we conducted a deconvolution analysis using transcriptome data to infer immune cells signatures and investigate their associations with LF (Appendix Fig. S4). The results were largely in line with those from TME gene expression signatures. LFs were also related to WGS-based DNA mutational signatures, including those implicated in endogenous (i.e., clock-like:SBS1, *APOBEC*:SBS13) and exogenous (e.g., tobacco smoke:SBS4, DBS2) exposures, as well as signatures with unknown aetiology (i.e., SBS40a,b,c, ID5) (Fig. 2C). LFs were also related to the presence of ccRCC somatic cancer driver mutations, of which associations with *PBRM1, SETD2, BAP1*, and *TP53* were replicated in the validation phase (Figs. 2C and EV3C).

In the following sections, we explored the relationship between the LF and ccRCC aetiology and progression features by triangulating evidence from the multi-layered molecular annotations and epidemiological data. We focused on the LF with shared sources of inter-patient variation across different omics data and associated with aetiological risk factors (Table 1; Table EV5).

## The major ccRCC molecular component is linked to cellular mitotic age

LF1 was robustly associated with chronological age (Table 1). This major component of ccRCC inter-patients' variation had large contribution from DNA methylation data (Fig. 1A). LF1 was typified by the DNA hypermethylation of CpG islands annotated to functional regions of the genome (Figs. 2A and EV3A). Of the differential methylated regions associated with LF1, the CpG sites annotated to *ZNF471* displayed the highest correlation with its transcript levels ($r = 0.70$, $P = 3.4 \times 10^{-19}$) (Dataset EV1). Such DNA methylation changes have been noted in cellular ageing processes (Bell et al, 2019; Marttila et al, 2015). LF1 levels were also associated with clock-like DNA mutational signatures (i.e., SBS1). Together, this prompted us to explore this ccRCC component in the context of biological age.

Biological age is a complex process that reflects an individual's physiological state over time; varying aspects can be measured by epigenetic clocks (Rutledge et al, 2022). LF1 was correlated with a range of epigenetic clocks (Fig. EV4), showing a remarkable correlation with the age-adjusted mitotic-like clock epiTOC2 ($r \geq 0.89$, $P < 0.001$). This epigenetic clock explained around 80% of the variance in LF1 across ccRCC tumours (Fig. 3A). EpiTOC2 is based on the CpG sites of Polycomb target genes that are unmethylated at birth but become progressively methylated as cells replicate, thus proposed to represent cellular mitotic age (Teschendorff, 2020). Consistent with this, pathway analysis of gene expression levels correlated with LF1 included pathways enriched for Polycomb (*EZH2*) target genes ($P = 8.3 \times 10^{-05}$; Table EV4). LF1 was also associated with mitotic clocks derived from different omics data, such as clock-like DNA mutational signature (SBS1) (Discovery: $P = 1.7 \times 10^{-04}$, Fig. 2C; Validation: $P = 2.1 \times 10^{-04}$; Fig. EV3C), as mentioned above, but also WGS-derived telomere attrition (Discovery: $P = 0.015$, Validation: $P = 0.019$; Dataset EV2). It was also associated with the accumulation of somatic copy number alterations (Discovery: $P = 8.4 \times 10^{-04}$, Validation: $P = 0.040$; Dataset EV2) and the presence of somatic cancer driver mutations in chromatin remodelling genes *PBRM1* and *SETD2* ($P < 0.001$; Figs. 2C and EV3C; Dataset EV2). LF1 was also associated with tumour stage and grade (Table 1; Fig. 3B), as well as patient survival (Fig. EV2). Patients in the top quintile of LF1 were estimated to be 23 times more likely to be late-stage tumours (III and IV) and 9 times more likely to be high-grade tumours (grade 3–4), compared to those in the bottom quintile (Dataset EV2). Nevertheless, LF1 varied within tumour stage and grade categories (Fig. 3B) and multivariate analysis suggested that the associations between LF1 and its key molecular findings did not appear to be driven by tumour stage and grade alone (Dataset EV2). Interestingly, LF1 levels were partially explained by previously described kidney cancer DNA methylation clusters (Ricketts et al, 2018) ($r^2 = 56\%$, Fig. 1B). Both shared common features, such as the hypermethylation of functional regions in the genome, enrichment for somatic driver mutations (*SETD2* and *PBRM1*), and associations with tumour stage, grade, and overall survival. Nevertheless, multivariable regression analyses showed that LF1 levels could provide additional information in predicting these features when compared to previously described ccRCC methylation clusters in the validation set (TCGA-KIRC, Table EV6).

**Table 1.  Association between latent factors 1–6 and epidemiological data of ccRCC patients.**

**Discovery set (N = 151)**

| | Factor 1 Beta ± SE | P value | Factor 2 Beta ± SE | P value | Factor 3 Beta ± SE | P value | Factor 4 Beta ± SE | P value | Factor 5 Beta ± SE | P value | Factor 6 Beta ± SE | P value |
|---|---|---|---|---|---|---|---|---|---|---|---|---|
| **Social-demographic variables** | | | | | | | | | | | | |
| Age at diagnosis | **0.17 ± 0.09** | **0.045** | 0.00 ± 0.08 | 0.959 | 0.00 ± 0.08 | 0.990 | 0.01 ± 0.08 | 0.908 | 0.06 ± 0.09 | 0.480 | 0.02 ± 0.09 | 0.772 |
| Sex (male) Ref=Female | 0.17 ± 0.17 | 0.316 | **0.47 ± 0.17** | **0.006** | −0.01 ± 0.16 | 0.971 | −0.32 ± 0.17 | 0.229 | 0.10 ± 0.17 | 0.586 | **−0.43 ± 0.17** | **0.013** |
| **Prognostic variables** | | | | | | | | | | | | |
| Stage (III+IV/late) Ref=I+II/early | **0.79 ± 0.17** | **1.04E-05** | **0.65 ± 0.17** | **2.61E-04** | 0.10 ± 0.17 | 0.568 | 0.15 ± 0.18 | 0.423 | **0.64 ± 0.18** | **5.13E-04** | −0.10 ± 0.18 | 0.602 |
| Grade (III+IV/high) Ref=I+II/low | **0.51 ± 0.17** | **0.002** | **0.63 ± 0.16** | **9.90E-05** | 0.08 ± 0.16 | 0.608 | −0.08 ± 0.17 | 0.641 | **0.67 ± 0.16** | **6.94E-05** | −0.02 ± 0.17 | 0.929 |
| **Main risk factors** | | | | | | | | | | | | |
| Body mass index (kg/m²) | −0.15 ± 0.08 | 0.073 | −0.08 ± 0.08 | 0.337 | 0.01 ± 0.08 | 0.873 | 0.00 ± 0.08 | 0.954 | −0.14 ± 0.08 | 0.105 | 0.07 ± 0.08 | 0.430 |
| Hypertension (yes) Ref=No | −0.10 ± 0.18 | 0.577 | −0.10 ± 0.18 | 0.552 | −0.11 ± 0.17 | 0.508 | 0.15 ± 0.18 | 0.398 | 0.03 ± 0.18 | 0.873 | 0.01 ± 0.18 | 0.968 |
| Diabetes (yes) Ref=No | −0.14 ± 0.25 | 0.570 | −0.25 ± 0.24 | 0.306 | −0.01 ± 0.23 | 0.975 | 0.15 ± 0.24 | 0.529 | 0.34 ± 0.25 | 0.167 | 0.13 ± 0.25 | 0.598 |
| Family history of RCC (yes) Ref=No | 0.61 ± 0.37 | 0.101 | −0.21 ± 0.37 | 0.559 | −0.52 ± 0.35 | 0.143 | −0.46 ± 0.37 | 0.207 | −0.03 ± 0.38 | 0.946 | −0.03 ± 0.37 | 0.927 |
| Tobacco smoking (ever) Ref=Never | 0.04 ± 0.17 | 0.812 | 0.26 ± 0.16 | 0.123 | −0.21 ± 0.16 | 0.189 | −0.05 ± 0.17 | 0.762 | 0.32 ± 0.17 | 0.064 | **0.73 ± 0.16** | **8.73E-06** |
| Alcohol drinking (ever) Ref=Never | 0.13 ± 0.21 | 0.535 | −0.15 ± 0.21 | 0.462 | −0.48 ± 0.20 | 0.017 | −0.09 ± 0.21 | 0.655 | 0.16 ± 0.22 | 0.469 | −0.07 ± 0.21 | 0.748 |
| **Kidney function-related variables** | | | | | | | | | | | | |
| Creatinine (mg/dL) | 0.01 ± 0.09 | 0.935 | 0.07 ± 0.09 | 0.455 | 0.12 ± 0.09 | 0.158 | −0.08 ± 0.09 | 0.330 | −0.03 ± 0.09 | 0.781 | −0.19 ± 0.09 | 0.038 |
| Cystatin C (mg/dL) | 0.10 ± 0.09 | 0.278 | 0.02 ± 0.09 | 0.782 | −0.03 ± 0.09 | 0.713 | 0.05 ± 0.09 | 0.608 | 0.02 ± 0.09 | 0.844 | 0.11 ± 0.09 | 0.254 |

**Validation sets (N = 324/TCGA-KIRC; N = 462/IARC ccRCC[b])**

| | Factor 1[a] Beta ± SE | P value | Factor 2[b] Beta ± SE | P value | Factor 3[b] Beta ± SE | P value | Factor 4[b] Beta ± SE | P value | Factor 5[b] Beta ± SE | P value | Factor 6[a] Beta ± SE | P value |
|---|---|---|---|---|---|---|---|---|---|---|---|---|
| **Social-demographic variables** | | | | | | | | | | | | |
| Age at diagnosis | **0.23 ± 0.05** | **2.00E-05** | | | | | | | | | | |
| Sex (male) Ref=Female | | | **0.49 ± 0.09** | **3.70E-07** | | | | | | | **−0.55 ± 0.11** | **9.64E-07** |
| **Prognostic variables** | | | | | | | | | | | | |
| Stage (III+IV/late) Ref=I+II/early | **0.89 ± 0.10** | **5.81E-18** | **0.45 ± 0.09** | **2.31E-06** | | | | | **0.50 ± 0.10** | **3.39E-07** | | |
| Grade (III+IV/high) Ref=I+II/low | **0.68 ± 0.10** | **7.50E-11** | **0.47 ± 0.10** | **1.36E-06** | | | | | **0.58 ± 0.10** | **5.34E-09** | | |
| **Main risk factors** | | | | | | | | | | | | |
| Tobacco smoking (ever) | | | | | −0.01 ± 0.10 | 0.943 | | | | | 0.13 ± 0.09 | 0.158* |
| Alcohol drinking (ever) | | | | | | | | | | | | |

[a]ccRCC tumour samples from TCGA-KIRC cohort; [b]ccRCC tumour samples from TCGA-KIRC studies with available transcriptome data; Baseline model: Factor ~ variable + sex (whenever possible) + age at diagnosis (whenever possible) + country (whenever possible); estimates of continuous variables (factors, age at diagnosis, and body mass index) were represented as value per standard deviation. Binary variables were categorised as follows: ever/never (tobacco smoking and alcohol drinking), yes/no (hypertension, diabetes, and family history of kidney cancer/RCC), III + IV/I + II (pathological tumour grade and stage); country variable not applicable to TCGA-KIRC since all cases from USA. *Limited information in the validation set related to tobacco smoking (TCGA-KIRC: 39 never, 30 ever smokers, 255 missing); more details about frequencies of these variables in Table EV1. Tumour stage and tumour grade were grouped into two categories due to the small sample size by strata in the discovery set (see Table EV1). Replicated results were highlighted in bold font.

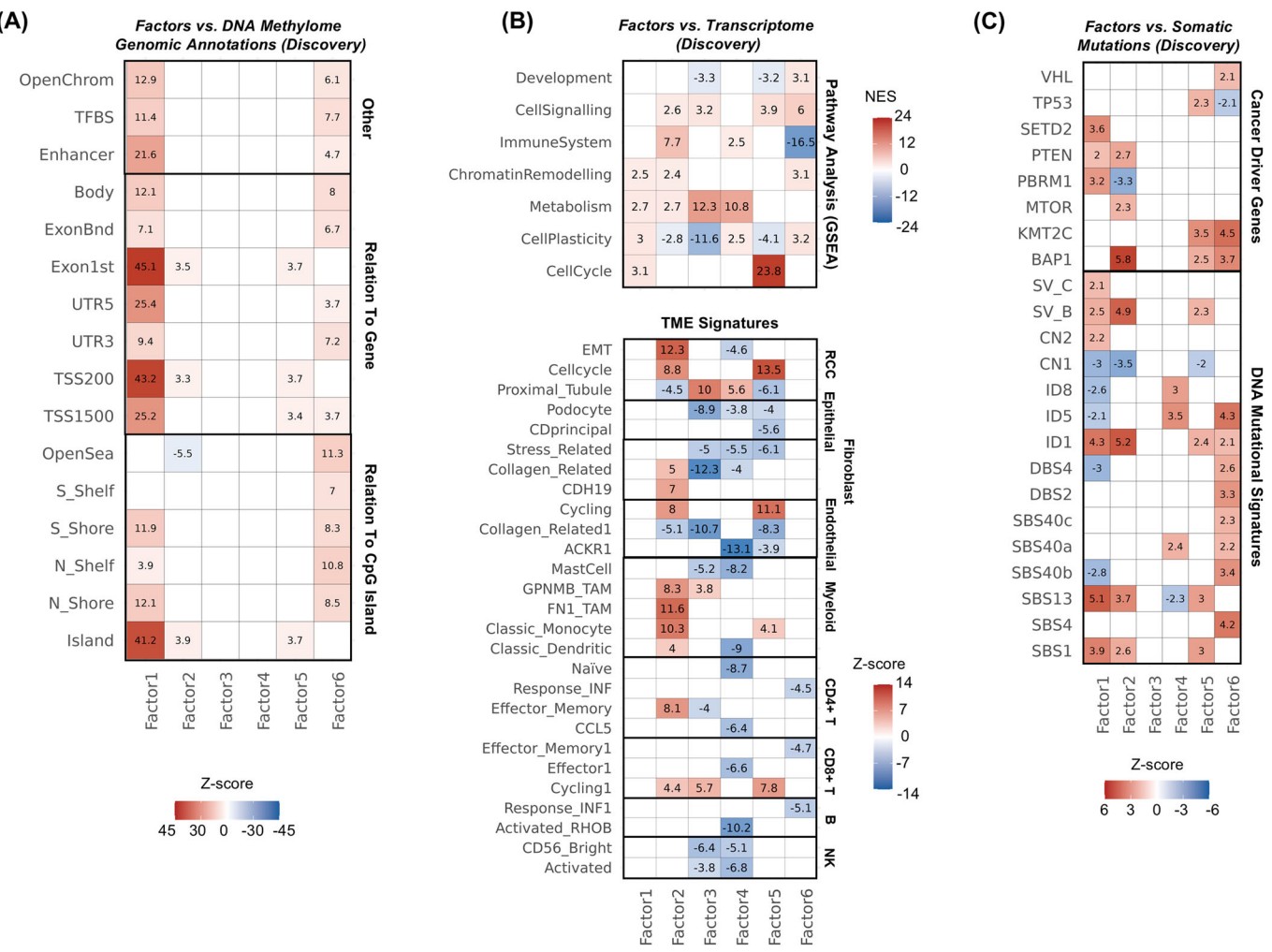

**Figure 2.  Associations between latent factors and molecular features of ccRCC tumours in the discovery set.**

Heatmaps showing the Z-scores (beta divided by standard error) of linear regression analyses between latent factors (outcome) and molecular features related to the three omics layers included in the Multi-Omics Factor Analysis (MOFA), adjusting by age at diagnosis and sex. (A) For the DNA methylome layer ($N = 120$), the average beta methylation levels of the 5000 MOFA CpG sites by genomic annotation related to CpG island (island, shores, shelves, and open sea), gene (proximal/TSS200 and distal/TSS1500 promoters, UTRs, exons, and body), and other regulatory regions (open chromatin, transcription factor binding site/TFBS, and enhancer) were used as predictors. (B) For the transcriptome layer, pathway analysis was conducted on the top 500 genes correlated with each latent factor using the GSEA database and fgsea R package (v1.27.1). The top 10 enriched biological pathways for each latent factor (FDR < 0.05) were manually categorised into functional groups based on their descriptions in the GSEA database and the canonical functions of the associated genes, including development, cell signalling, immune system, chromatin remodelling, metabolism, cell plasticity, and cell cycle. We then summed the normalised enrichment scores (NES), provided by the pathway analysis, by functional category to represent the major biological processes enriched for each latent factor. For the tumour microenvironment (TME) gene expression signatures, it was shown the statistically significant associations between latent factors 1 to 6 and the 27 representative TME signatures in ccRCC tumours. The association estimates were derived from the analyses in the validation sets (IARC ccRCC series: 462 tumours for LF 2–5; TCGA-KIRC: 323 tumours for LF1 and 6) after adjustments by covariates (sex, age at diagnosis, and country of origin whenever possible). Associations that passed multiple-testing correction (false discovery rate/FDR < 0.05, 162 tests) were represented. The associations were represented as Z-scores (beta divided by standard error; Z-scores >0 in shades of red; Z-score <0 in shades of blue). The ccRCC tumour microenvironment signatures (CD4 + T, B, NK, endothelial, myeloid, CD8 + T, epithelial and fibroblast cells) and kidney cancer meta programmes/RCC (epithelial-to-mesenchymal transition/EMT and cell cycle) were derived from previous published single-cell RNA sequencing data (Li et al, 2022). (C) The somatic mutation profile based on whole-genome sequence data (WGS, $N = 151$) was represented by ccRCC driver mutations (binary; presence or absence) and DNA mutational signatures (continuous). Regression models included age at diagnosis, sex, and country of origin as covariates. Values represented as shades of red (Z-scores >0) and blue (Z-score <0). The associations that passed multiple-testing correction (FDR < 0.05) within each group of variables were represented. Tobacco-related (SBS4, DBS2), clock-like (SBS1, ID1), *APOBEC* (SBS13), copy number (CN) and structural variation (SV) DNA mutational signatures. Source data are available online for this figure.

Consistent with the notion that cellular mitotic age is accelerated in tumour tissues, the patient's tumour material had marked higher values of LF1 (tumours: median of $0.43 \pm 0.95$; normal tissue: median of $-0.87 \pm 0.20$, $P = 1.6 \times 10^{-45}$) than paired normal kidney tissue after adjustments for chronological age

(Fig. 3C). Taken together, the main source of ccRCC inter-patients' variability is related to cellular mitotic age.

The LF1 signature was inferred across tumour samples from the TCGA pan-cancer cohorts ($N = 8040$) explaining on average 15% of the variance in tumour DNA methylation data across TCGA cohorts

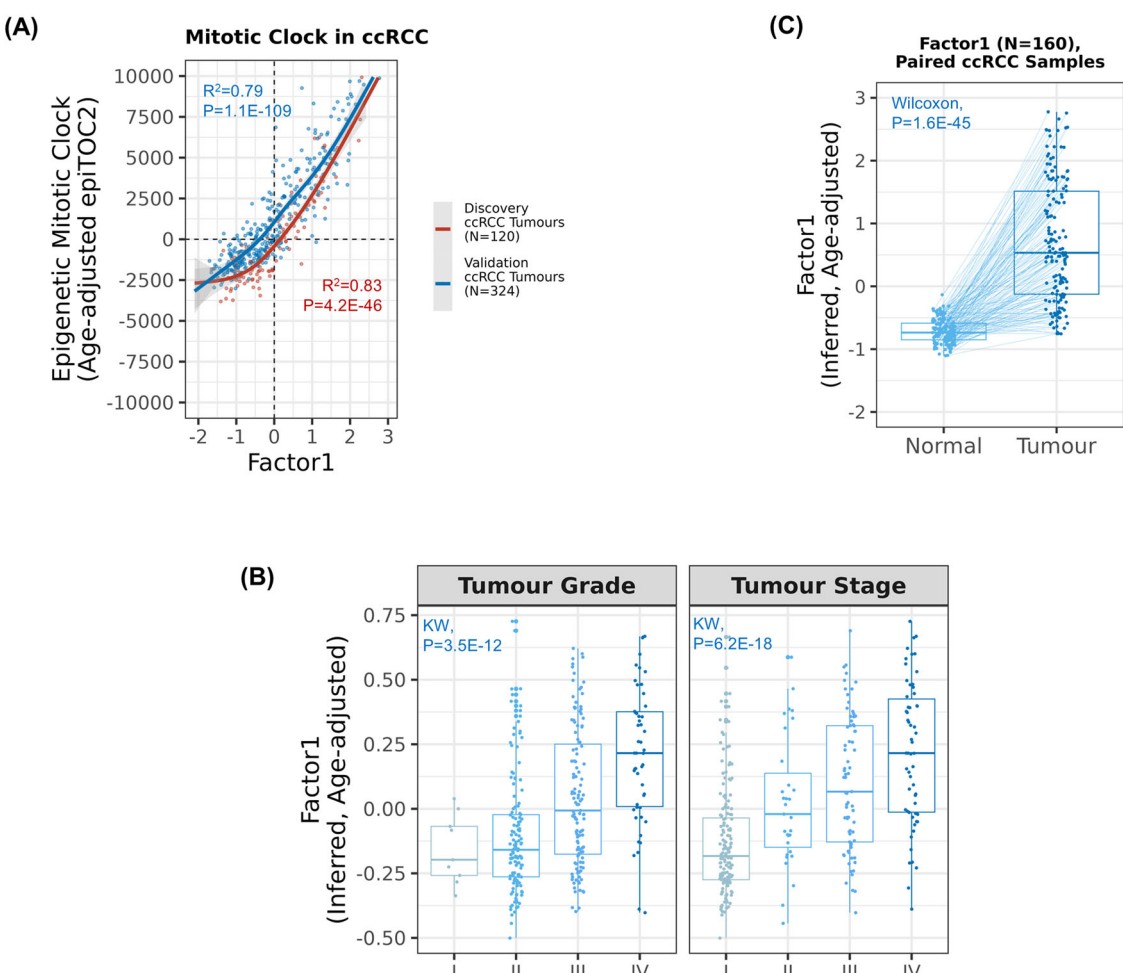

**Figure 3. Relationship between latent factor 1, the mitotic-like epigenetic clock epiTOC2, and prognosis.**

Analyses were performed using the residuals of the cellular mitotic age epigenetic clock epiTOC2 after adjusting by chronological age. (A) Smooth regression lines between latent factor 1 and the age-adjusted epiTOC2 (Discovery (red dots): 120 tumours, Validation/TCGA-KIRC (blue dots): 324 tumours) in ccRCC tumours. *P* values derived from linear regression models (Factor 1 ~age-adjusted epiTOC2). (B) Age-adjusted residuals of latent factor 1 across different ccRCC tumour stages ($N = 322$) and grades ($N = 320$) from TCGA validation set ($N = 323$). Statistical comparison between medians was performed using Kruskal-Wallis's test. (C) Comparison of medians of paired normal adjacent kidney tissues (light blue) and ccRCC tumours (dark blue) for age-adjusted latent factor 1 (TCGA-KIRC: 160 normal-tumour pairs). Lines connect matched samples. *P* values from Wilcoxon signed-rank test. The boxplots depict the distribution of the data with the central line representing the median (50th percentile). The bounds of the box correspond to the interquartile range (IQR), extending from the 25th percentile (lower quartile) to the 75th percentile (upper quartile). The whiskers show the minimum and maximum values within 1.5 times the IQR from the quartiles, while data points beyond the whiskers are considered outliers.

*P* values < 0.05 were considered statistically significant. Observed latent factor 1 used for the regression models in the discovery set while the latent factor 1 signature was used for the analyses in the validation set.

(Dataset EV3). Our pan-cancer analyses showed that LF1 signature was associated with the patient's chronological age ($P = 8.1 \times 10^{-40}$), epiTOC2 ($P < 1.0 \times 10^{-307}$), dosage of the clock-like mutational signature SBS1 ($P = 2.8 \times 10^{-15}$), WGS-telomere length ratio ($P = 0.022$), copy number alterations ($P = 1.0 \times 10^{-05}$), and the presence of *PBRM1* ($P = 1.3 \times 10^{-10}$) and *SETD2* ($P = 2.2 \times 10^{-09}$) somatic cancer driver genes, independent to ccRCC tumours (TCGA-KIRC) (Table 2). Intriguingly, these effects also appeared particularly prominent in other histological subtypes of kidney cancer (TCGA-KICH/chromophobe, TCGA-KIRP/papillary), including the prognostic value of LF1 in patients with papillary kidney cancer (TCGA-KIRP: $P = 6.8 \times 10^{-17}$) (Dataset EV3).

LF1 was weakly correlated with LF5 ($r = 0.25$; $P = 0.005$; Appendix Fig. S1) and associated with SBS1 (Discovery:

$P = 0.003$, Fig. 2C; Validation: $P = 0.010$, Fig. EV3C). By contrast, LF5 was reproducibly associated with SBS13/*APOBEC* activity (Discovery: $P = 0.027$, Fig. 2C; Validation: $P = 1.4 \times 10^{-04}$, Fig. EV3C) and not associated with chronological age ($P = 0.480$) (Table 1). LF5 was associated with the presence of somatic cancer driver mutations in *TP53* (Discovery: $P = 0.025$; Validation: $P = 6.0 \times 10^{-10}$), as well as with changes in the expression of cell cycle genes and cycling cells (all $P < 0.001$; Figs. 2B,C and EV3B,C). LF5 was also associated with prognostic variables, such as tumour stage and grade (all $P < 0.001$; Table 1), and patient overall survival (Fig. EV2). When comparing LF5 results (expression of cell cycle genes and *TP53* mutations) with previous studies, they resembled those found in the molecular proliferative subgroup of ccRCC patients reported by a previous study (Motzer et al, 2020), both of

Table 2. Pan-cancer associations between latent factors 1, 2 and 6 and molecular and epidemiological features across TCGA cohorts (excluding TCGA-KIRC).

| Feature | Beta | SE | P value | R² (Omic) |
|---|---|---|---|---|
| **Latent factor 1 (mitotic clock)** | | | | |
| SBS1 ($N = 7464$) | 0.095 | 0.012 | 2.84E-15 | 15% DNA methylome |
| Age-adjusted epigenetic mitotic clock (epiTOC2) ($N = 7721$)* | 0.766 | 0.007 | <1.00E-307 | |
| WGS-Telomere length ratio, log2(tTL:nTL) ($N = 588$) | −0.101 | 0.044 | 0.022 | |
| Altered copy number fraction ($N = 7141$) | 0.061 | 0.014 | 1.03E-05 | |
| *PBRM1* (Ref = 'Absence') ($N = 7317$) | 0.507 | 0.079 | 1.35E-10 | |
| *SETD2* (Ref = 'Absence') ($N = 7317$) | 0.397 | 0.066 | 2.25E-09 | |
| Age at diagnosis ($N = 7721$) | 0.179 | 0.013 | 8.06E-40 | |
| **Latent factor 2 (immune cells and epithelial–mesenchymal transition/EMT)** | | | | |
| EMT signature ($N = 9284$)* | 0.484 | 0.010 | <1.00E-307 | 5% Transcriptome |
| Cell cycle signature ($N = 9284$) | 0.139 | 0.011 | 2.57E-37 | |
| Fibronectin1-tumour-associated macrophage (FN1_TAM) Signature ($N = 9284$)* | 0.489 | 0.010 | <1.00E-307 | |
| *IL20RB* (DNA methylation levels) ($N = 7707$) | −0.051 | 0.016 | 0.002 | |
| *IL20RB* (gene expression levels) ($N = 9284$) | −0.018 | 0.015 | 0.223 | |
| *WT1* (gene expression levels) ($N = 9284$) | 0.094 | 0.016 | 4.98E-09 | |
| *BAP1* (Ref = 'Absence') ($N = 7816$) | 0.221 | 0.094 | 0.019 | |
| Sex (Ref = 'Female') ($N = 8299$) | 0.051 | 0.028 | 0.074 | |
| **Latent factor 6 (GSTP1 metabolism and interferon-gamma response)** | | | | |
| *GSTP1* (DNA methylation levels) ($N = 7721$) | 0.317 | 0.018 | 4.37E-68 | 6% DNA methylome |
| *GSTP1* (gene expression levels) ($N = 7680$) | −0.223 | 0.020 | 6.20E-28 | |
| Tobacco signature (DNAm) ($N = 7721$) | 0.368 | 0.014 | 6.28E-150 | |
| Total mutation burden ($N = 6752$) | −0.025 | 0.013 | 0.054 | |
| Interferon-gamma responsive CD4 + T signature ($N = 7680$) | −0.302 | 0.011 | 4.01E-161 | |
| Sex (Ref = 'Female') ($N = 7763$) | −0.005 | 0.028 | 0.865 | |
| Tobacco smoking (Ref = 'Never Smoker') ($N = 2590$) | −0.057 | 0.046 | 0.219 | |

The joint model also excluded TCGA-SKCM since most of the tumour samples were derived from metastatic sites while the other cancer sites were from primary solid tumours. Factors were inferred using the most informative features of the observed factors selected by the LASSO regression model. The LM model: Factor ~ molecular features + age at diagnosis + TCGA-cohort. Beta estimates represented one unit per s.d. *Value < 1E-307, which is the lowest float number that could be represented.

which associated with poor overall survival of patients. Combined, these results suggested that LF5 may represent actively proliferating cells in ccRCC tumours through *TP53* somatic cancer driver mutations, with impact on patient's prognosis.

## Relationship between LF2 and EMT

LF2 was associated with DNA methylation changes (annotated to CpG island and CpG in open seas), immune system-related pathways, and *BAP1* somatic cancer driver mutations (all $P < 0.001$; Figs. 2 and EV3). Higher LF2 levels were observed in male patient's tumours compared with female patient's tumours (Table 1), late-stage, high-grade tumours (Table 1; Fig. 4B), and it was associated with worse patient survival (Fig. EV2). This ccRCC component was associated with the presence of TME signatures related to EMT process, cycling endothelial cells, cell cycle kidney meta pro-grammes, and myeloid cells (particularly fibronectin-positive tumour-associated macrophages/FN1_TAM) (all $P < 0.001$; Figs. 2B and EV3B).

We performed a differentially methylated region analysis to identify the genomic regions associated with LF2 levels (Dataset EV4). Among the regions loaded with LF2 levels, the ones showing the highest correlations between DNA methylation and RNA levels mapped to EMT genes (*IL20RB*: $r = -0.84$, $P = 4.2 \times 10^{-33}$; *KRT19*: $r = -0.71$, $P = 1.2 \times 10^{-19}$; *WT1*: $r = -0.70$, $P = 7.1 \times 10^{-19}$) (Dataset EV4). As LF2 levels increase, the regulatory regions of these genes lose DNA methylation, leading to a corresponding upregulation in their expression (Fig. 4A). As *BAP1* somatic mutations were also associated with LF2 levels, we evaluated if this epigenetic effect could be modulated by *BAP1* somatic cancer driver mutations. Indeed, this epigenetic effect was particularly pronounced for *IL20RB* and *WT1* in the presence of *BAP1* somatic cancer driver mutations (Fig. EV5). As LF2 levels were related to stage IV ccRCC tumours (Fig. 4B) and EMT, we evaluated if LF2 and EMT genes were related to metastasis. Higher levels of LF2 were associated with the likelihood of ccRCC patients having distant metastasis ($P = 0.007$). This relationship with metastasis was also found with *WT1* ($P = 0.017$) and *IL20RB* ($P = 7.4 \times 10^{-06}$) expression in ccRCC

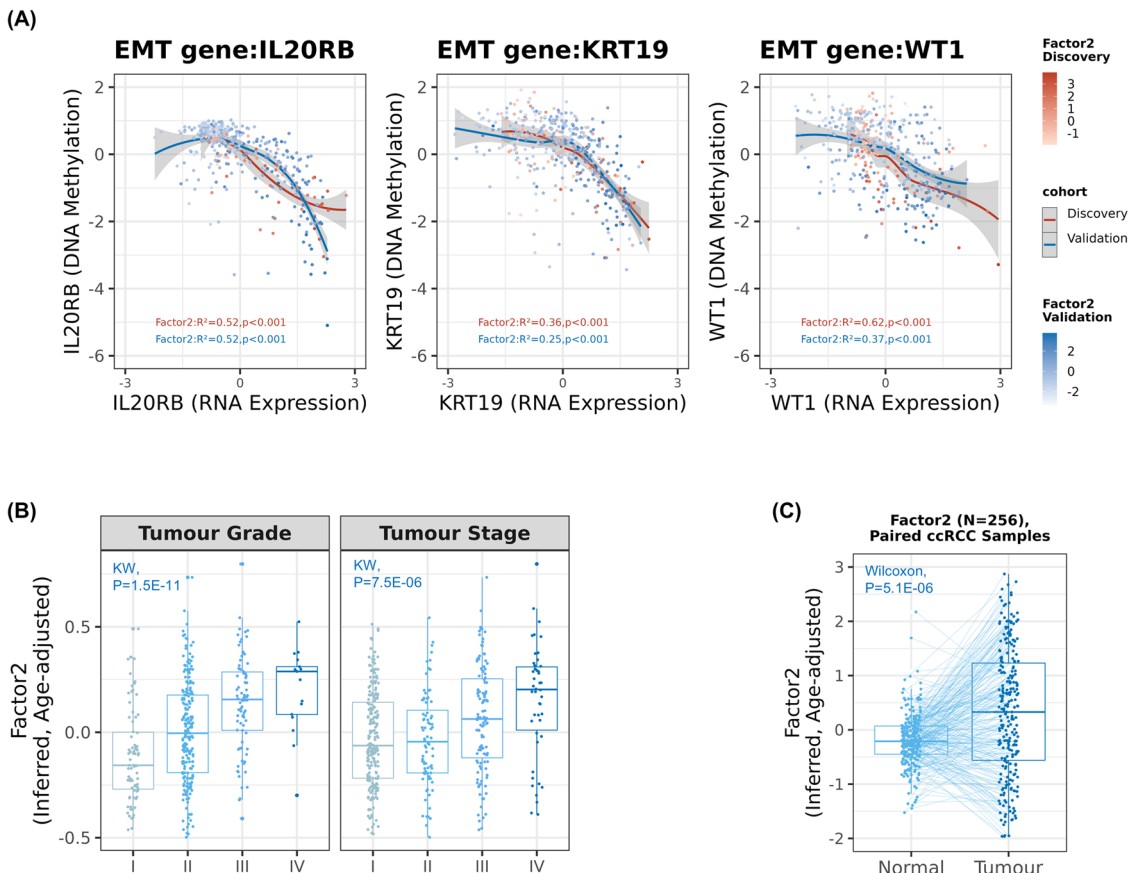

**Figure 4. Relationship between latent factor 2 and epithelial–mesenchymal transition process (EMT).**

(A) Smooth regression lines between the average DNA methylation and RNA levels of EMT genes (*IL20RB*, *KRT19*, and *WT1*). Each dot was coloured by the values of latent factor 2 in the discovery (shades of red; IARC ccRCC: 120 tumours) and validation sets (shades of blue; TCGA-KIRC: 323 tumours). The proportion of variance of latent factor 2 explained by both DNA methylation and RNA levels of each gene was calculated by linear regression models. The regression model provided adjusted $R^2$ values and $P$ values (Discovery: $P_{IL20RB} = 2 \times 10^{-16}$ (DNAm) and $3 \times 10^{-18}$ (RNA), $P_{KRT19} = 5 \times 10^{-8}$ (DNAm) and $1 \times 10^{-15}$ (RNA), $P_{WT1} = 6 \times 10^{-25}$ (DNAm) and $1 \times 10^{-9}$ (RNA); Validation: $P_{IL20RB} = 2 \times 10^{-29}$ (DNAm) and $1 \times 10^{-49}$ (RNA), $P_{KRT19} = 5 \times 10^{-8}$ (DNAm) and $9 \times 10^{-26}$ (RNA), $P_{WT1} = 3 \times 10^{-32}$ (DNAm) and $2 \times 10^{-16}$ (RNA)). (B) Residuals of latent factor 2 signature after adjusting by chronological age were plotted across different ccRCC tumour stages (322 tumours) and grades (320 tumours) from TCGA validation set (323 tumours). $P$ values from Kruskal–Walis's test. (C) Comparison of paired normal adjacent kidney tissues (light blue) and ccRCC tumours (dark blue) for age-adjusted latent factor 2 (IARC ccRCC validation set: 256 normal-tumour pairs). Lines connect matched samples. The boxplots depict the distribution of the data with the central line representing the median (50th percentile). The bounds of the box correspond to the interquartile range (IQR), extending from the 25th percentile (lower quartile) to the 75th percentile (upper quartile). The whiskers show the minimum and maximum values within 1.5 times the IQR from the quartiles, while data points beyond the whiskers are considered outliers. P values from Wilcoxon signed rank. $P$ values < 0.05 were considered statistically significant.

tumours (Table EV7). In relation to previous findings, LF2 features partially overlapped with those of two molecular subgroups of ccRCC patients (T-effector/proliferative and stromal/proliferative) identified in a previous study (Motzer et al, 2020), both of which associated with poor survival. This overlap included a higher frequency of *BAP1* somatic cancer driver mutations, striking intratumorally adaptive immune responses, and enrichment for gene expression related to the EMT pathway. However, we additionally identified the relationship between the epigenetic regulation of EMT genes and *BAP1* somatic cancer driver mutations in ccRCC.

In line with previous works showing an important immune infiltrate in ccRCC tumours, the levels of LF2 varied within normal kidney tissues, with higher levels in matched ccRCC tumours (tumours: median of $0.21 \pm 1.28$; normal tissue: median of $-0.21 \pm 0.53$, $P = 5.1 \times 10^{-06}$) (Fig. 4C).

When considering other tumour types in a pan-cancer analysis, LF2 was also associated with the EMT ($P < 1.0 \times 10^{-307}$) and FN1_TAM signatures ($P < 1.0 \times 10^{-307}$), as well as *WT1* expression ($P = 5.0 \times 10^{-09}$) across tumour types, independent to ccRCC tumours (Table 2, Dataset EV5). On the other hand, the association with *BAP1* somatic cancer driver mutations ($P = 0.019$) and *IL20RB* expression ($P = 0.223$) appeared to be ccRCC specific (Table 2; Dataset EV5). Together, LF2 seemed to capture multiple biological processes involved on EMT and tumour progression.

Since LF2 along with LF1 and LF5 levels were associated with patient overall survival, we generated Cox Proportional-Hazards models to evaluate how they compared to existing models of patient prognosis based on tumour stage, grade and somatic mutations (*BAP1*, *SETD2*, and *PBRM1*) (Table EV8). The combined LF model outperformed the ccRCC driver mutation model (Validation: C-Index=$0.71 \pm 0.03$ vs. $0.63 \pm 0.03$, $P_{diff} = 0.033$) while displaying

statistically similar performance to the clinical (C-Index=0.73 ± 0.03, $P_{diff}$ = 0.560) and integrated models (Validation: C-Index=0.75 ± 0.03, $P_{diff}$ = 0.215). Given the increment of 2–3% in the performance of the integrated model in relation to the clinical one, our results suggest that these LFs have the potential to complement tumour stage and grade in predicting ccRCC patient prognosis.

## LF6, GSTP1 metabolism, and interferon-gamma response

LF6 was related to the DNA hypermethylation of CpG sites in proximity to CpG islands and gene bodies (Figs. 2A and EV3A). The DNA methylation changes correlated to LF6 with the highest functional impact on transcript levels were annotated to the xenobiotic metabolism gene *GSTP1* (glutathione S-transferase pi) ($r = -0.71$, $P = 2.4 \times 10^{-19}$; Dataset EV6). Higher levels of LF6 were related to the DNA hypermethylation of its promoter region (Discovery: $P = 9.0 \times 10^{-13}$; Validation: $P = 1.5 \times 10^{-05}$) and decreased expression of *GSTP1* (Discovery: $P = 5.6 \times 10^{-08}$; Validation: $P = 3.2 \times 10^{-05}$) (Fig. 5A), estimated to jointly explain around 36% of the variance in LF6. This source of inter-patient variation was associated with tobacco smoking (Table 1), the environmental exposure robustly associated with ccRCC risk by observational studies (Hsieh et al, 2017) and dosage of the tobacco-related DNA mutational signatures (SBS4: $P = 5.5 \times 10$; DBS2: $P = 0.001$) in the discovery set (Fig. 2C). It was also associated with the tobacco smoking DNA methylation signature (Discovery: $P = 0.003$; Validation: $P = 0.010$; Fig. 5A) and total tumour mutation burden (Discovery: $P = 0.003$; Validation: $P = 0.029$; Fig. 5A), particularly among tobacco smokers ($P_{Ever\ Smk} = 4.3 \times 10^{-04}$ vs. $P_{Never\ Smk} = 0.487$; Table EV9), as well as other DNA mutational signatures frequently found in kidney tumours (SBS40b: $P = 0.001$, ID5: $P = 2.6 \times 10^{-05}$; Fig. 2C). ccRCC tumours of female patients had higher loadings of this ccRCC component (Table 1). ccRCC tumours with higher loadings of LF6 displayed downregulation of immune system pathways, particularly lower levels of the gene expression signatures related to interferon-gamma (*INFG*) response (all $P < 0.001$; Figs. 2B and EV3B; Table EV4). Interestingly, LF6 was significantly higher in a subset of normal kidney tissues in comparison with the matched ccRCC tumours (median of 0.95 ± 0.26 vs. median of −0.47 ± 0.89, $P = 6.1 \times 10^{-56}$) (Fig. 5B).

In pan-cancer analysis using TCGA cohorts (Dataset EV7), we observed associations between LF6 and DNA methylation ($P = 4.3 \times 10^{-68}$), RNA levels ($P = 6.2 \times 10^{-28}$) of *GSTP1* as well as *INFG* response ($P = 4.0 \times 10^{-161}$) across different tumour types.

## Discussion

Our study demonstrates the advantages of an integrative multi-omics approach, as emphasised by others (Argelaguet et al, 2018; Cantini et al, 2021). By integrating transcriptome, DNA methylation, and somatic mutation data, we were able to enhance the resolution of molecular structures within patient tumours beyond what single-omics methods could achieve. More importantly, this comprehensive strategy allowed us to extract complementary layers of information, enabling a deeper triangulation of biological meaning in relation to disease mechanisms. We identified key sources of inter-patient variation that not only complemented but also extended

previous findings from clustering analyses (Ricketts et al, 2018; Thorsson et al, 2018). For example, LF1, a structure highly correlated with a previously described ccRCC methylation feature (Ricketts et al, 2018), gained added significance in our study when we linked it to the WGS-derived somatic mutation signature (SBS1) and its association with chronological age. This connection led us to explore the hypothesis that LF1 may be related to biological ageing. By integrating such findings with epidemiological and genomic annotations, we gained novel insights into their biological relevance, adding both depth and interpretability to our results.

Our approach identified two sources of variance related to biological ageing. The largest source of variance between ccRCC tumours was related to cellular mitotic age, which is a measure of cellular proliferative history. Our study is the first to describe the relationship between different types of mitotic clocks measured across omics layers—namely DNA methylation (epiTOC2), somatic mutations (SBS1), and telomere length in the same tumour sample. The association between LF1 and epiTOC2 was particularly strong. The fact that LF1 levels, which are derived from an unsupervised approach, covaried with epiTOC2 is striking, and it reinforces the hypothesis that the DNA methylation changes at CpG sites of Polycomb target genes might be particularly useful in representing cellular mitotic age (Teschendorff, 2020). LF1 levels were associated with somatic cancer driver mutations in *PBRM1* and *SETD2*, chromatin remodelling genes linked to cell senescence (Lee et al, 2016) and proliferation (Cai et al, 2019; Dominguez et al, 2016), which appears consistent with our observation that higher mitotic rates are related to these features. Similarly, LF1 was related to late-stage, high-grade tumours, again consistent with the expected higher mitotic rates (Hakimi et al, 2013; Motzer et al, 2020; Ricketts et al, 2018). Cancer risk factors can also affect mitotic rate. For example, higher mitotic rates are observed in histologically normal lung epithelial tissue of tobacco smokers compared to non-smokers (Young et al, 2018). In this study, the LFs were derived from ccRCC tumours, and thus, the effects of carcinogenesis on the mitotic rate may mask the effects of potential exposures. Exploring hypothesis that consider how mitotic rates in the histologically normal kidney are affected by risk factors like tobacco use, obesity, or acute kidney injury (Kellum et al, 2021) appears warranted. Interestingly, we also distinguished between cellular mitotic age (LF1) and the active proliferation state of the cell (LF5). The latter may represent a more aggressive and proliferation-centric tumour phenotype through *TP53* mutations affecting directly the cell cycle, while the former could affect indirectly the mitotic rate by regulating the chromatin state through *PBRM1* and *SETD2* and thereby facilitating access to transcription factors.

ccRCC tumours are characterised by important infiltration of immune cells (Li et al, 2022; Ricketts et al, 2018; Şenbabaoğlu et al, 2016; Thorsson et al, 2018). Accordingly, our multi-omics approach identified sources of inter-patient variation associated with different aspects of TME in ccRCC tumours (LF2-LF6). Of those, LF2 levels were strongly correlated with EMT, which is an important process in cancer invasiveness (Hanahan, 2022). We observed epigenetic upregulation of EMT-related genes, particularly in tumours harbouring *BAP1* somatic cancer driver mutations, which are linked to advanced-stage tumours, distant metastasis, and poorer prognosis. The top gene associated with LF2 was *IL20RB*, which we found to be epigenetically upregulated in ccRCC tumours. This mechanism was previously identified by others conferring worse prognosis of ccRCC patients (Guo et al, 2022;

**(A)**

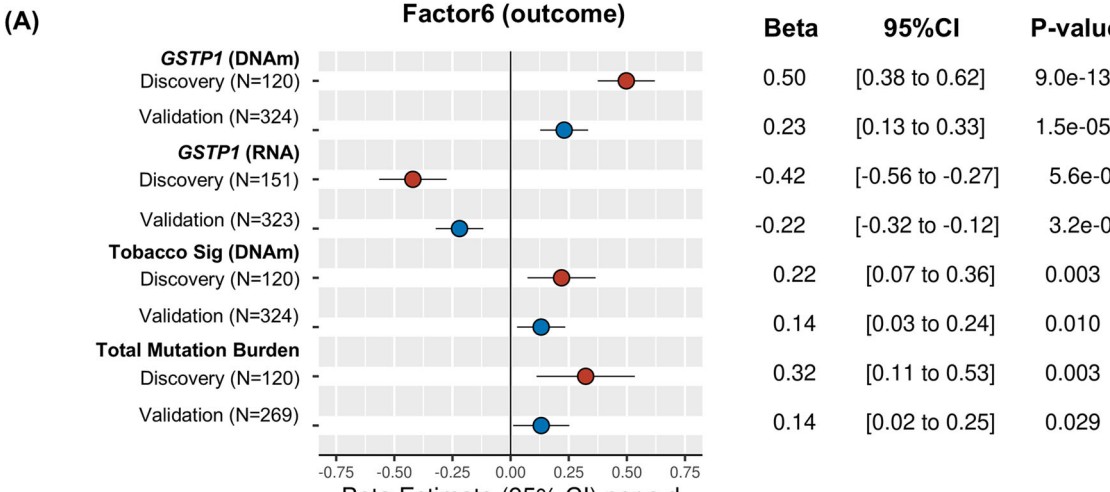

| | Beta | 95%CI | P-value |
|---|---|---|---|
| **GSTP1 (DNAm)** | | | |
| Discovery (N=120) | 0.50 | [0.38 to 0.62] | 9.0e-13 |
| Validation (N=324) | 0.23 | [0.13 to 0.33] | 1.5e-05 |
| **GSTP1 (RNA)** | | | |
| Discovery (N=151) | -0.42 | [-0.56 to -0.27] | 5.6e-08 |
| Validation (N=323) | -0.22 | [-0.32 to -0.12] | 3.2e-05 |
| **Tobacco Sig (DNAm)** | | | |
| Discovery (N=120) | 0.22 | [0.07 to 0.36] | 0.003 |
| Validation (N=324) | 0.14 | [0.03 to 0.24] | 0.010 |
| **Total Mutation Burden** | | | |
| Discovery (N=120) | 0.32 | [0.11 to 0.53] | 0.003 |
| Validation (N=269) | 0.14 | [0.02 to 0.25] | 0.029 |

**(B)**

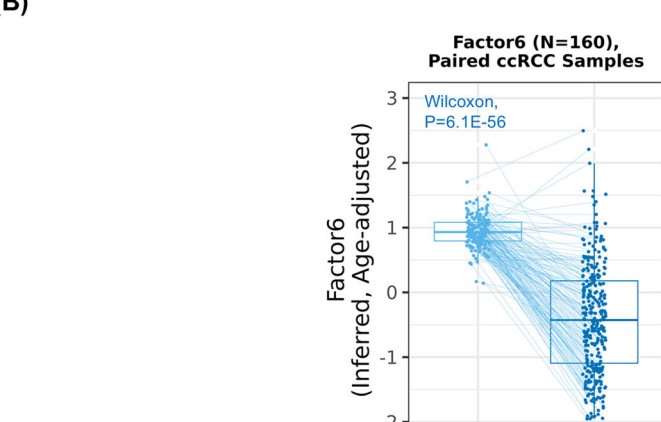

**Figure 5. Associations between latent factor 6 and molecular features related to exogenous exposures in ccRCC tumours.**

(A) Forest plot showing the results of multivariable regression analyses of latent factor 6 (outcome) and *GSTP1* methylation (Average m-values of CpG sites annotated to *GSTP1*, DNAm, continuous) and gene expression levels (RNA, continuous), and DNA methylation signature of tobacco smoking trained to predict self-reported tobacco smoking status (5 CpG sites, continuous, epiTob), and total mutation burden (whole-genome for discovery and whole-exome for validation) in both discovery (120 tumours for DNA methylation and 151 for gene expression data) and validation (TCGA-KIRC; 324 tumours; 268 cases with total mutation burden analysis) sets. Covariates used in the regression models were sex, age at diagnosis, and country of origin (whenever possible). For analyses of total mutation burden, ccRCC cases from Romania ($N = 31$) were excluded from the discovery set. Beta estimates were represented as an increase in the effect of the selected features per 1 unit of standard deviation increase in latent factor 6. Blue dots when discovery and red dots when validation ccRCC tumour cohorts. *P* values < 0.05, derived from the linear regression models, were considered statistically significant. Observed latent factor 6 used for the regression models in the discovery set while a signature was used for the analyses in the validation set. (B) Comparison of paired normal adjacent kidney tissues (light blue) and ccRCC tumours (dark blue) for age-adjusted latent factor 6 (TCGA-KIRC: 160 normal-tumour pairs). Lines connect matched samples. The boxplots depict the distribution of the data with the central line representing the median (50th percentile). The bounds of the box correspond to the interquartile range (IQR), extending from the 25th percentile (lower quartile) to the 75th percentile (upper quartile). The whiskers show the minimum and maximum values within 1.5 times the IQR from the quartiles, while data points beyond the whiskers are considered outliers. *P* values from the Wilcoxon signed-rank test *P* values < 0.05 were considered statistically significant.

Laskar et al, 2021). Additionally, LF2 showed a strong correlation with pro-inflammatory macrophage signatures. Given that *IL20RB* overexpression activates pro-inflammatory macrophages in lung tissue (Zhu et al, 2024), and that LF2 was associated with macrophage signatures here, we hypothesize that *IL20RB* over-expression may similarly drive macrophage activation in ccRCC tumours and, in conjunction with EMT processes, contribute to the aggressive tumour characteristics observed. We also identified the epigenetic upregulation of *WT1*, a gene essential for kidney

development that controls the balance between epithelial and mesenchymal states. Intriguingly, *WT1* acts as a tumour suppressor in Wilms tumours—a rare form of childhood kidney cancer—through loss-of-function mutations and epigenetic silencing (Miller-Hodges and Hohenstein, 2012) while it is also suggested to act as oncogene in several tumour types in adults (Sugiyama, 2001; Yang et al, 2007). Our results suggest that the epigenetic activation is a major contributor to *WT1*'s oncogenic role in ccRCC by promoting EMT. Notably, the epigenetic effect in *WT1* appeared

to be orientated towards the gene body rather than its promoter region. Overall, LF2 may represent the relationship between EMT in the renal epithelial cells and TME, a phenomenon particularly pronounced in ccRCC tumours with *BAP1* mutations. This finding suggests the dual role of *BAP1* somatic mutations—not only as drivers involved in ccRCC initiation but also as enhancers of EMT-related epigenetic regulation, contributing to disease progression.

LF6 emerged as a novel source of inter-patient variation in ccRCC linked to tobacco smoke exposure. This was primarily driven by the epigenetic silencing of *GSTP1*, a detoxifying enzyme that normally conjugates glutathione to harmful genotoxic compounds, enhancing their solubility and facilitating excretion (Schnekenburger et al, 2014). The loss of *GSTP1*, primarily through epigenetic silencing, has been reported in various cancers (Cui et al, 2020; Rønneberg et al, 2008; Su et al, 2007), and could make proximal tubule cells in the kidney, which metabolise these compounds, more vulnerable to genotoxic stress (Young et al, 2018). Consistent with this notion, LF6 was associated with molecular marks of tobacco smoke exposure, including SBS4 and epigenetic marks of tobacco exposure, in patients with a history of tobacco smoking. LF6 was also associated with DNA mutational signatures where the mechanism is unknown (SBS40 components, DBS4, and ID5), suggesting that genotoxic compounds may be related to their aetiology. Additionally, LF6 was linked to immune cell responses, specifically through *INFG* signalling, which plays a dual role in cancer, exerting both tumour-suppressing and tumour-promoting effects by influencing immune and stromal cells (Scholz et al, 2021). In kidney tumours, *INFG* was described as a gatekeeper of disease progression by restraining the clonal expansion of cancer cells (Young et al, 2018). We found that higher LF6 levels correlated with a reduction in *INFG*-responsive immune gene expression, both in ccRCC and in other cancer types. This suggests an immune-suppressive environment, possibly allowing for unchecked tumour growth. In summary, LF6 appears to reflect both impaired detoxification due to *GSTP1* loss and altered immune regulation via *INFG*, creating a permissive environment for tumour progression.

Some limitations of this study should be recognised. First, integrative multi-omics approaches are designed to capture the principal sources of inter-patient variation, which means that minor sources of variation may not be well-resolved. This could explain why some aspects, such as the DNA mutation signature SBS22 found in Romanian patients (15% of the total), and proposed to be caused by aristolochic acid exposure, were not associated with LF in the discovery set. Second, our integrative model was trained on tumour tissue data, making it more representative of attributes related to tumour progression, such as tumour stage and grade, rather than the physiological states related to aetiology, which may be better represented from the histologically normal material. We used a discovery and validation approach to limit how the play of chance might influence our results and their subsequent interpretation. Nevertheless, differences between the two sets posed limitations. First, the LF in the validation series were inferred rather than measured directly. The lack of independent sets for training, testing, and validation, ideal for the development of a signature (or biomarker), may have reduced the accuracy of the signature inference and, therefore, the model's ability to capture associations. Similarly, differences in omics data, which include DNA methylation (850k vs. 450k), transcriptome (array *vs*. RNA sequencing) and somatic mutation profile (WGS vs. WXS, particularly for DNA mutational signature attribution), might have further contributed

to differences in resolution. Indeed, while most features we highlighted here were replicated, some were less pronounced in the validation series, which may be related to this limitation. Similarly, epidemiological data were only partially recorded in the validation cohort; for instance, tobacco information was missing in 81% of patients. Thus, the lack of association with certain metrics should be interpreted cautiously.

In conclusion, our study has explored how the underlying biological processes involved in the initiation and progression of ccRCC tumours by unravelling molecular marks linked to endogenous and exogenous exposures detected at different omics layers. This includes ccRCC components linked to cellular mitotic age, tobacco smoke exposure, EMT processes and the tumour microenvironment. Further exploration of these findings may lead to novel intervention targets based on the description of disease aetiology and potential biomarkers with prognostic value for patients.

## Methods

**Reagents and tools table**

| Reagent/resource | Reference or source | Identifier or catalogue number |
| --- | --- | --- |
| **Experimental models** | | |
| Not applicable | | |
| **Recombinant DNA** | | |
| Not applicable | | |
| **Antibodies** | | |
| Not applicable | | |
| **Oligonucleotides and other sequence-based reagents** | | |
| Not applicable | | |
| **Chemicals, enzymes, and other reagents** | | |
| Not applicable | | |
| **Software** | | |
| Mutational Signature Analysis (MSA) v1.2 | https://gitlab.com/s.senkin/MSA | |
| SigProfilerMatrixGenerator v1.2.27 | https://github.com/AlexandrovLab/SigProfilerMatrixGenerator | |
| MOFA2, v1.10.0 | https://biofam.github.io/MOFA2/ | |
| R software v4.1.2 | https://www.r-project.org/ | |
| **Other** | | |
| EZ-96 DNA Methylation Kit | Zymo Research | ZD5004 |
| Infinium® Methylation EPIC | Illumina | WG-317-1003 |
| iScan system scanner | Illumina | |

## Participants of the study

### Discovery set

The participants included in the discovery set were part of the Mutographs project that was coordinated by the International

Agency for Research on Cancer (IARC/WHO) with available WGS, transcriptome (microarray), and DNA methylation data (newly generated in the current study) (Table EV1). The participants included in the study met the following criteria ($N = 151$): age at diagnosis >=18 years old (mean of $60.3 \pm 10.7$), reviewed diagnosis of primary ccRCC by pathologists following the guidelines from the International Cancer Genome Consortium, and no history of cancer treatment. The exclusion criteria were the non-availability of informed consent or suitable samples according to the protocol requirements. More details elsewhere (Senkin et al, 2024).

### Validation sets

To validate transcriptome findings, we used two IARC-led cohort studies described previously (Laskar et al, 2021). Both hospital-based studies where the transcriptome had been profiled for ccRCC tumour ($N = 462$), with a subset of matched normal adjacent kidney tissue samples ($N = 256$), used the same inclusion and exclusion criteria as the discovery set (Table EV1). To validate DNA methylation and somatic cancer driver mutations findings, the TCGA-KIRC cohort ($N = 324$) was used (TCGA, 2013). Pan-cancer analyses included 31 TCGA cohorts ($N = 8040$). The molecular and clinical information regarding the participants of TCGA cohorts are publicly available (https://portal.gdc.cancer.gov/) and DNA methylation data (normalised beta values), including primary tumours and matched normal adjacent tissues were obtained using TCGAbiolinks R package (version 2.22.3) (Colaprico et al, 2016).

## DNA mutational signatures and cancer driver mutations

DNA mutational signatures and cancer driver mutations were derived from WGS data from the Mutographs project, as described elsewhere (Senkin et al, 2024). Briefly, WGS was conducted on Illumina HiSeqX platform (Ilumina, San Diego, CA, USA) with a target coverage of 40× and sequence reads were aligned to GRCh38 human reference genome. Somatic variant calling was performed using the standard Wellcome Sanger Institute's analysis pipeline (https://github.com/cancerit). The activities of each DNA mutational signature were attributed along with the confidence intervals using the MSA tool (https://gitlab.com/s.senkin/MSA) (Senkin, 2021). For the identification of cancer driver mutations, dNdS approach restricting to a panel of known cancer genes (Martincorena et al, 2017) followed by a consensus annotation of candidate driver mutations using Cancer Gene Census (https://cancer.sanger.ac.uk/census) and Cancer Genome Interpreter (https://www.cancergenomeinterpreter.org) tools were used. To replicate the DNA mutational signatures-related findings in TCGA (WXS), mutation matrices were generated using SigProfilerMatrixGenerator (https://github.com/AlexandrovLab/SigProfilerMatrixGenerator) by applying the 'exome' option, which downsamples mutational matrices to the exome regions of the genome. Then, MSA tool was used to attribute the DNA mutational signatures based on COSMIC reference (V3.4) to explore their relationship with LFs in TCGA cohorts.

## Transcriptome data

Processed transcriptome data of normal adjacent kidney and ccRCC tumour samples used for both discovery and IARC validation sets were derived from previous studies (Laskar et al, 2021). Briefly, gene expression analysis was performed using Illumina HumanHT-12 v4 expression BeadChips (Ilumina, San

Diego, CA, USA), restricting to samples with RNA integrity >5. Raw probe intensities with signal-to-noise ratio >9.5 were further processed via variance-stabilising transformation and quantile normalisation using lumi package in R (v2.5). The probe sequences were aligned to the hg19 human reference genome. For downstream analyses, only probes with detection rate (quality metric) >5% in both paired normal and tumour samples were considered. Whenever multiple probes were mapped to a single gene, the probe with the highest detection rate was considered.

## DNA methylation profiling

The DNA methylation analyses of 121 ccRCC tumour samples were newly generated using Infinium Methylation EPIC (850 K) Bead-Chip (Illumina, San Diego, CA, USA) for the current study, as recently described elsewhere (Talukdar et al, 2021). Briefly, the DNA of samples underwent pre-processing steps as follows: bisulfite-conversion, whole-genome amplification, fragmentation, and hybridisation with complementary probe sequences on Bead-Chip. The images of the arrays were captured by iScan system scanner (Ilumina, San Diego, CA, USA) and probe intensities were obtained by GenomeStudio Software (Ilumina, San Diego, CA, USA). The processing steps of probes were performed using the implemented functions in methylkey R package (https://github.com/IARCbioinfo/methylkey). DNA methylation status was estimated by the β value—signal from the methylated probe divided by the overall signal intensity. The methylation levels of CpG sites were described as a continuous β value range between 0 (no methylation) and 1 (full methylation). Sample-specific quality controls were performed interrogating DNA methylation predicted sex and sample clustering based on the overall signal intensity median of the methylated and unmethylated channels. One low-quality sample was excluded from further analyses. β values were normalised using functional normalisation (FunNorm), and probes with missing rate >20% or flagged as 'crossReactive', 'SNP', and 'XY' were removed. We used the SVA package (v3.35.2) to remove any potential batch effects from the DNA methylation data, ensuring that any biases introduced by conducting the experiments in two batches did not confound our analysis or influence the results. For regression purposes, β-values were converted to M-values (Talukdar et al, 2021). DNA methylation sites were annotated with the information provided by Illumina and the University of California Santa Cruz (UCSC) database (hg19).

## MOFA and LF signatures

MOFA was performed to integrate the different omics layers (DNA methylome, transcriptome, and somatic mutational profile from WGS) of overlapping ccRCC tumour samples (R package MOFA2, v1.10.0). We chose MOFA for our integrative multi-omics analysis because it supports different likelihood models, including both Gaussian (e.g., array transcriptomics) and non-Gaussian models (e.g., presence or absence of driver mutations and DNA mutational signatures), and it also tolerates missing values (Argelaguet et al, 2018). This flexibility allows for a more comprehensive analysis of diverse data types compared to other omics integration tools (Cantini et al, 2021). DNA methylation data were missing for 30 out of the 151 tumour samples (discovery set) and they were imputed by MOFA, as previously described (Argelaguet et al, 2018).

As recommended by the software due to computational limitations, we selected the 5000 most variable features across samples for DNA methylome (CpG sites) and transcriptome data (gene expression levels) as continuous variables (Appendix Fig. S1). The somatic mutational profile derived from previous WGS (Senkin et al, 2024) was summarised as binary variables (presence or absence) of both ccRCC driver genes and DNA mutational signatures in different mutation contexts (SBS96, DBS72, ID83, CN48 and SV32), restricting to variables with more than five events in the discovery set. MOFA generated ten continuous orthogonal LF that explained important sources of variance across different omics data of ccRCC tumours (Appendix Fig. S1). Elbow statistic supported the selection of the first six LF for further analyses (Appendix Fig. S1D). No additional associations between LF 7–10 and epidemiological data were observed in the discovery set (Table EV5).

To infer an approximation of the LFs in the validation sets, we generated LASSO-based signatures using the most relevant layers (either DNA methylome or transcriptome) for each LF in the discovery set (Table EV2). The features correlated with each LF (FDR < 0.05 for 30,000 tests) were selected as variables for the LASSO regression models. The LASSO tune parameters were chosen by resampling the discovery set using the tidymodels metapackage in R (v1.0.0; wrapper of glmnet) that by default sets to ten the minimum number of features when the most stringent penalty is applied. The ten features selected by LASSO models were used to generate signatures that represented an approximation of each LF by adding up the scaled values of the normalised $m$-values (DNA methylation) or log2-transcripts per million (transcriptome) of each feature multiplied by the respective LASSO regression coefficients (Table EV2). These signatures were then used to infer the LF across TCGA cohorts, IARC normal, and tumour ccRCC samples (validation sets). The gene expression signatures for factors 2–5, calculated initially using transcriptome data (IARC micro-array), were also applied to voom-transformed RNA sequencing data (Law et al, 2014) of TCGA cohorts.

### Additional molecular annotations

Global DNA methylation levels were estimated using the mean $m$-values of locus-specific repetitive elements (*LINE1* and *Alu*) across chromosomes using REMP package in R (v1.24.0), since these repetitive elements were reported to be more accurate in estimating global DNA methylation levels than averaging the methylation levels of all CpG sites from 850 K/450k Illumina arrays (Lisanti et al, 2013; Zheng et al, 2017). DNA methylation-based clocks were calculated using methylclock (v1.6.0) and dnaMethyAge (v0.1.0) packages in R. The inference of TME cells was conducted using pre-defined lists of genes identified in a previous single-cell RNA sequencing study focused on ccRCC tumours (Li et al, 2022). We chose to use these gene lists as they were derived from ccRCC tumours. These gene lists were derived from the Seurat scRNA-seq integration pipeline to integrate ccRCC tumour cells from different patients. Briefly, non-negative factorisation analysis was used to cluster ccRCC tumour cells, and gene expression profiles of each cluster were determined. Then, these clusters were assigned to specific cell types or major biological programmes within ccRCC based on curated gene sets from external databases. We restricted our analyses to the cell signatures in which more than 75% of genes identified in the original study (Li et al, 2022) were

also present in our study after quality control. The scRNA-derived signatures were correlated within the bulk transcriptome data of the patient's tumours (Appendix Fig. S3) and 27 representative signatures ($r < 0.80$ within the original TME cell type or cell programme) could be derived from the 65 TME signatures. The TME signatures were estimated by the sum of the scaled gene expression value of each gene belonging to a signature by sample. We additionally performed a deconvolution analysis using quantiseqr (v1.12.0) package in R to estimate the proportions of immune cells based on the transcriptome data of ccRCC tumours. The DNA methylation signature for tobacco smoking status (epiTob) was calculated by summing the methylation levels of the 5 CpG sites (cg05575921, cg26703534, cg23480021, cg08118908, cg00336149) up associated with self-reported smoking status (Chamberlain et al, 2022). Molecular variables from TCGA cohorts were also included in the current study, such as the homologous recombination DNA repair deficiency score (Thorsson et al, 2018), the copy number alteration fraction of the genome (Knijnenburg et al, 2018), telomere length ratio based on WGS data (Barthel et al, 2017) and copy number variation DNA mutational signatures (Steele et al, 2022). The total mutation burden in the discovery cohort was estimated, excluding hypermutated cases from Romania (Senkin et al, 2024).

### Statistical analyses between LFs and epidemiological data

Linear regression models between LFs (outcome) and epidemiological data (predictors) were performed, and associations were adjusted, whenever possible, by age at diagnosis, sex, and country of origin. Cox proportional hazard regression analysis was used to assess overall patient survival. All statistical analyses were performed using R tools (V4.1.2).

## Data availability

DNA methylation data (850k EPIC array) of 121 ccRCC tumours are available here (850k EPIC array: NCBI/GEO database GSE269180). R code used in the current work can be found on the project webpage at: https://github.com/ricardocortezcardoso/multi_omic_code.

The source data of this paper are collected in the following database record: biostudies:S-SCDT-10_1038-S44320-024-00072-3.

## Peer review information

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

## Acknowledgements

The authors would like to acknowledge Mutographs consortium, TCGA Research Network (https://www.cancer.gov/tcga), K2 and CCE studies, and the contribution of specimen donors and research groups involved in these resources. This work was funded as part of the Mutographs team supported by the Cancer Grand Challenges partnership funded by Cancer Research UK (C98/A24032). Funding for gene expression data was provided by the US National Institutes of Health (NIH), National Cancer Institute (U01CA155309). Institut National du Cancer funded the DNA methylation data (GeniLuc2017-1-TABAC-03-CIRC-1-[TABAC17-022]).

## Author contributions

**Ricardo Cortez Cardoso Penha**: Conceptualisation; Data curation; Formal analysis; Investigation; Visualisation; Methodology; Writing—original draft; Project administration; Writing—review and editing. **Alexandra Sexton Oates**: Formal analysis; Methodology; Writing—original draft; Writing—review and editing. **Sergey Senkin**: Formal analysis; Methodology; Writing—original draft. **Hanla A Park**: Writing—review and editing. **Joshua Atkins**: Writing—original draft; Writing—review and editing. **Ivana Holcatova**: Resources; Writing—review and editing. **Anna Hornakova**: Resources; Writing—review and editing. **Slavisa Savic**: Resources; Writing—review and editing. **Simona Ognjanovic**: Resources; Writing—review and editing. **Beata Świątkowska**: Resources; Writing—review and editing. **Jolanta Lissowska**: Resources; Writing—review and editing. **David Zaridze**: Resources; Writing—review and editing. **Anush Mukeria**: Resources; Writing—review and editing. **Vladimir Janout**: Resources; Writing—review and editing. **Amelie Chabrier**: Methodology; Project administration. **Vincent Cahais**: Formal analysis; Methodology. **Cyrille Cuenin**: Methodology. **Ghislaine Scelo**: Resources; Funding acquisition; Writing—review and editing. **Matthieu Foll**: Writing—original draft; Writing—review and editing. **Zdenko Herceg**: Writing—original draft; Writing—review and editing. **Paul Brennan**: Resources; Writing—review and editing. **Karl Smith-Byrne**: Writing—original draft; Writing—review and editing. **Nicolas Alcala**: Methodology; Writing—original draft; Writing—review and editing. **James D Mckay**: Conceptualisation; Resources; Formal analysis; Supervision; Funding acquisition; Validation; Investigation; Methodology; Writing—original draft; Project administration; Writing—review and editing.

Source data underlying figure panels in this paper may have individual authorship assigned. Where available, figure panel/source data authorship is listed in the following database record: biostudies:S-SCDT-10_1038-S44320-024-00072-3.

## Disclosure and competing interests statement

The authors declare no competing interests. Where authors are identified as personnel of the International Agency for Research on Cancer/World Health Organization, the authors alone are responsible for the views expressed in this article and they do not necessarily represent the decisions, policies, or views of

the International Agency for Research on Cancer/World Health Organization. This study was conducted in accordance with the ethical principles outlined in the Declaration of Helsinki. Informed consent was obtained for all participants included in the discovery and validation sets. Ethical approvals were obtained from Local and Federal Research Ethics Committees, and from the IARC Ethics Committee (IEC Project 17-10A4). For the TCGA datasets, also used as validation set, the enrolling, collection, clinical and genomic data processing and distributions are subject to 45-CFR-46 (the "Common Rule") governing protection of human research subjects. Under the revised TCGA consent policy, re-consent of still-living participants is no longer a programme-imposed requirement. The Project Team described the best practices for informed consent for participating in TCGA in this memo (http://cancergenome.nih.gov/abouttcga/policies/informedconsent).

# Expanded View Figures

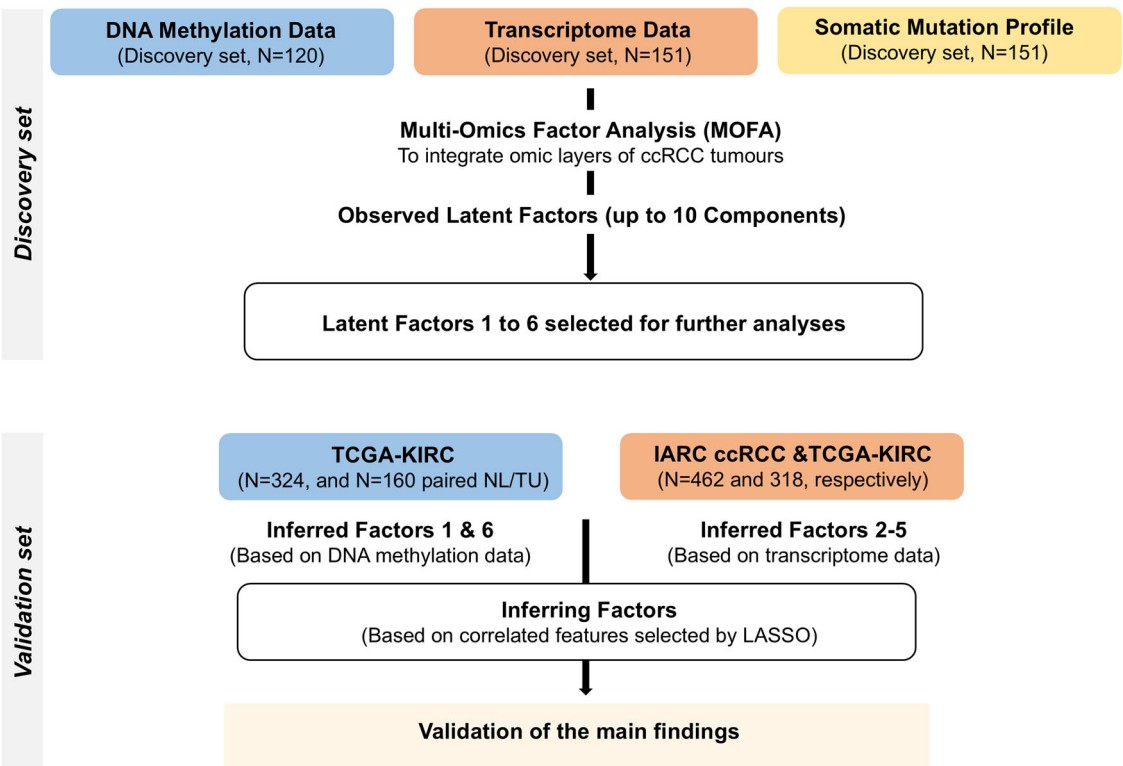

**Figure EV1. Study design.**

(Upper) For the discovery set (IARC series), we used DNA methylation (120 tumours), transcriptome (151 tumours), and somatic mutation profile derived from whole-genome sequencing data of the overlapping clear cell renal cell carcinoma (ccRCC) tumour samples (151 tumours) to perform a Multi-Omics Factor Analysis (MOFA) to uncover the sources of inter-patient variation in the ccRCC data. For these analyses, the top 5000 most variable features within DNA methylation (continuous variables; M-values) and transcriptome data (continuous variables; log2-transcripts per million) along with DNA mutational signatures and cancer driver mutations (binary variables; presence or absence) derived from whole-genome sequencing were used as inputs to MOFA. The output of MOFA was 10 orthogonal latent factors. The first six latent factors were selected using the elbow method. In parallel, no additional associations between latent factors 7 to 10 and epidemiological data were observed (more details in Table EV5). (Bottom) For validation purposes, we used Least Absolute Shrinkage and Selection Operator (LASSO) regression models to select the most informative independent features correlated with each latent factor, CpG sites or gene expression levels since these two omics layers accounted for over 90% of inter-patient's variability in the discovery set, and based on them, we calculated signatures to infer the latent factors in tumour and paired normal adjacent/tumour samples in two independent datasets (TCGA-KIRC: 324 and 160 normal-tumour pairs; IARC ccRCC series: 462 tumours).

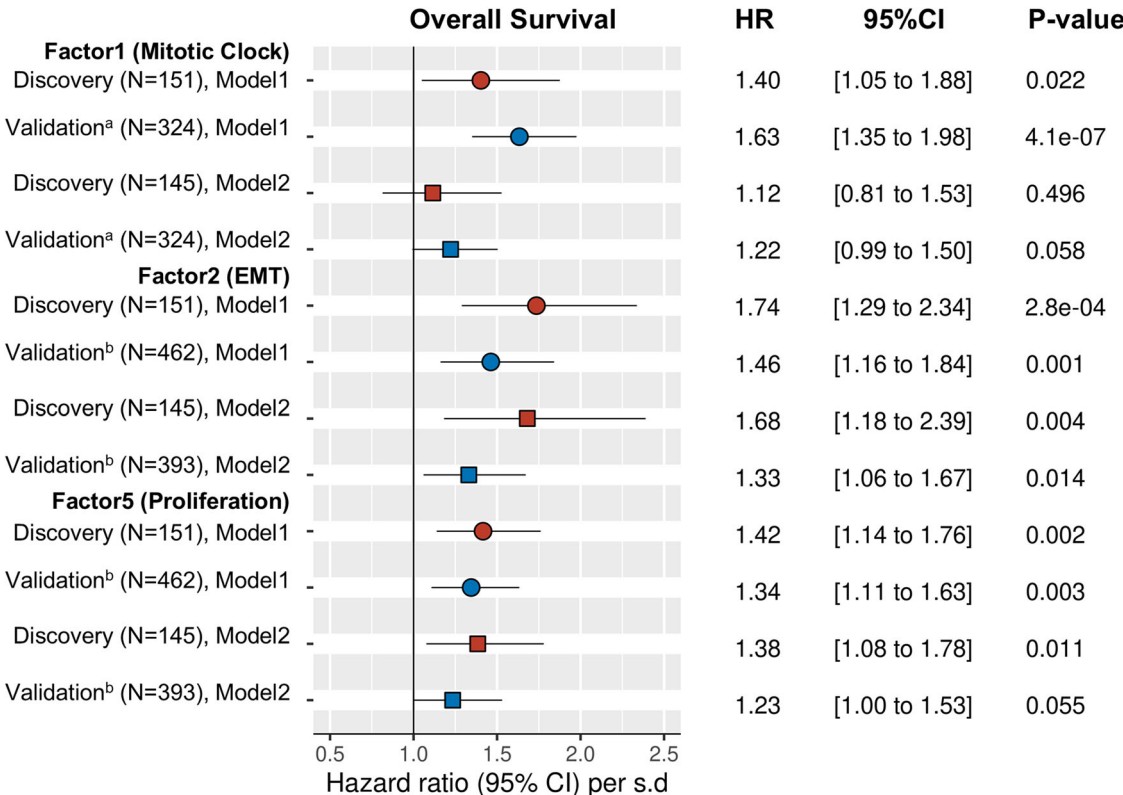

**Figure EV2. Molecular components associated with prognosis of ccRCC patients.**

Cox proportional-hazards models for assessing overall survival of ccRCC patients in relation to latent factor 1 (mitotic-like epigenetic clock epiTOC2), 2 (epithelial–mesenchymal transition/EMT), and 5 (cell cycle), adjusting for age at diagnosis, sex (model 1; circle shape), and additionally by tumour stage and grade (I + II vs. III + IV; model 2; square shape) in the discovery (red) and validation (blue) datasets. Hazard ratios (HR) represented as an increase in relative mortality risk per 1 unit of standard deviation increase in factors. Two different validation sets were used according to the factors, TCGA-KIRC (a; 324 kidney tumour samples) for latent factor 1 and IARC series sets (b; 462 kidney tumour samples) for latent factors 2 and 5. Observed latent factors used for the regression models in the discovery set while signatures for the same latent factors were used for the analyses in the validation sets. *P* values < 0.05, derived from Cox proportional-hazards models, were considered statistically significant.

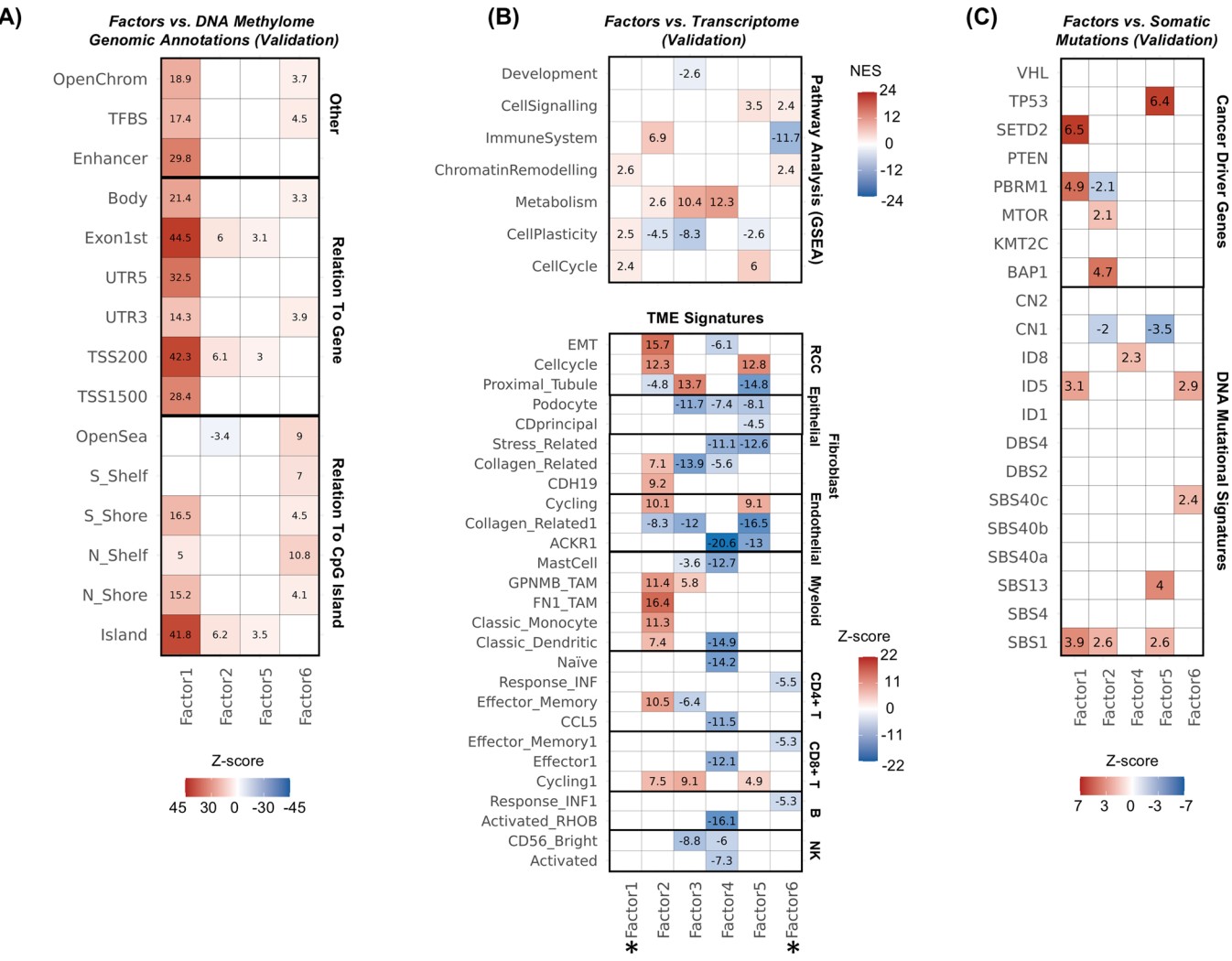

**Figure EV3. Validation of the associations between latent factors and molecular features of ccRCC tumours in the independent ccRCC datasets.**

For validation purposes, only the statistically significant associations in the discovery set were considered in the validation phase. Heatmaps showing the Z-scores (beta divided by standard error) of linear regression analyses between latent factor signatures (outcome) and genomic annotations and features related to the three omics layers, adjusting by age at diagnosis and sex. (A) For the DNA methylome layer (TCGA-KIRC: 324 tumours for DNA methylation-related latent factors 1 and 6; and 323 cases for the expression-related ones, latent factors 2–5), the average beta methylation levels of the 4,000 CpG sites that overlapped with discovery set were summarised by genomic annotations related to CpG island (island, shores, shelves, and open sea), gene (proximal/TSS200 and distal/TSS1500 promoters, UTRs, exons, and body), and other regulatory regions (open chromatin, transcription factor binding site/TFBS, and enhancer) and used as predictors. (B) For the transcriptome layer, pathway analysis was performed for the top 500 gene expression levels correlated with each latent factor using the GSEA database and fgsea R package (v1.27.1). Biological pathways present in the discovery set with $P$ value < 0.05 were manually categorised into functional groups based on their descriptions in the GSEA database and the canonical functions of the associated genes, including Development, Cell Signalling, Immune System, Chromatin Remodelling, Metabolism, Cell Plasticity, and Cell Cycle. We then summed the normalised enrichment scores (NES), provided by the pathway analysis, by functional category to represent the major biological processes enriched for each latent factor. For the tumour microenvironment (TME) gene expression signatures, it was shown the statistically significant associations between latent factors 1 to 6 and the 27 representative TME signatures in ccRCC tumours. The association estimates were derived from the analyses in the validation sets (IARC ccRCC series: 462 tumours for latent factors 2–5; TCGA-KIRC: 323 tumours for latent factors 1 and 6) after adjustments by covariates (sex, age at diagnosis, and country of origin whenever possible). The associations were represented as Z-scores (beta divided by standard error; Z-scores>0 in shades of red; Z-score<0 in shades of blue). The ccRCC tumour microenvironment signatures (CD4 + T, B, NK, endothelial, myeloid, CD8 + T, epithelial and fibroblast cells) and kidney cancer meta programmes/RCC (epithelial-to-mesenchymal transition/EMT and cell cycle) were derived from previous single-cell RNA sequencing data (Li et al, 2022). (C) The somatic profile based on whole-exome sequencing data (TCGA-KIRC: 268 tumours) was represented by ccRCC somatic driver mutations (binary; presence or absence) and DNA mutational signatures (continuous). Regression models included age at diagnosis, sex, and country of origin as covariates. Values represented as shades of red (Z-scores>0) and blue (Z-score<0). The associations with $P < 0.05$, derived from linear regression models, were represented. Tobacco-related (SBS4, DBS2), clock-like (SBS1, ID1), *APOBEC* (SBS13), copy number (CN) and structural variation (SV) DNA mutational signatures. Source data are available online for this figure.

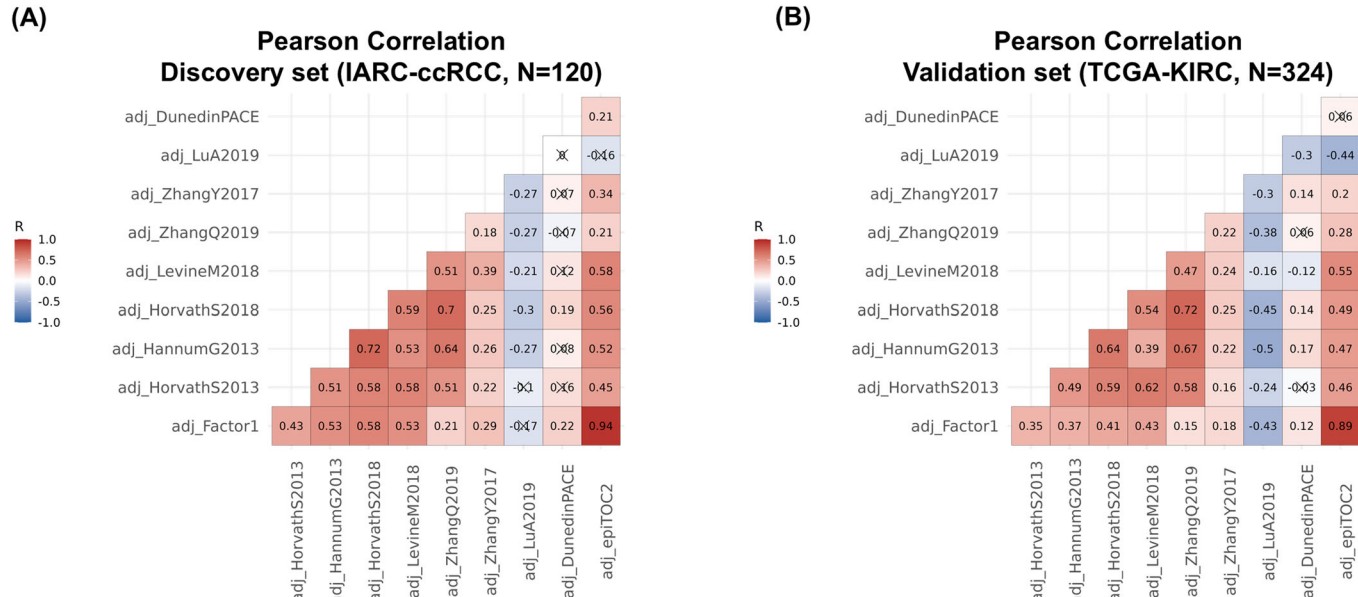

**Figure EV4. Correlation between latent factor 1 and epigenetic clocks.**

Heatmap represents the Pearson's correlation coefficients between the residuals of epigenetic clocks and latent factor 1 after adjusting by chronological age in ccRCC tumours from (**A**) IARC discovery (120 tumours) and (**B**) validation (TCGA-KIRC: 324 tumours) sets with DNA methylation data. Horvath's (Horvath et al, 2013, Horvath et al, 2018), Hannum's (Hannum et al, 2013), and Zhang's (Zhang et al, 2019) clocks trained on chronological age. EpiTOC2 is designed to predict mitotic cell rate. Zhang's (Zhang et al, 2017), Levine's (Levine et al, 2018), and DunedinPACE (Belsky et al, 2022) clocks to predict mortality risk. Lu's (Lu et al, 2019) to predict telomere length.

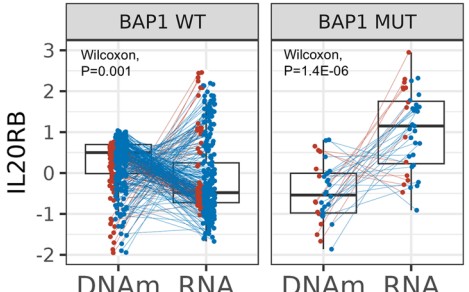

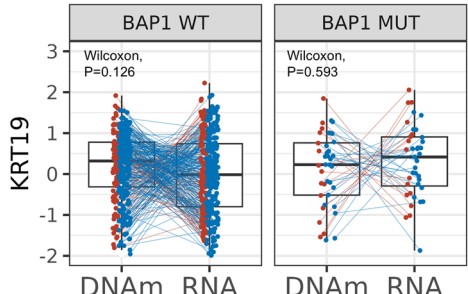

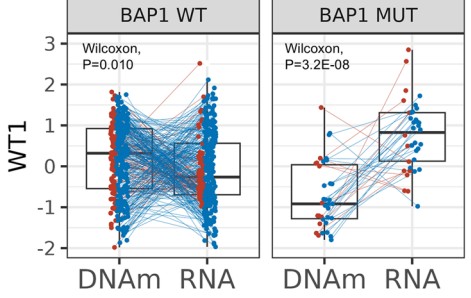

cohort

• Discovery

• Validation

**Figure EV5.  *BAP1* mutations and the epigenetic regulation of epithelial–mesenchymal transition (EMT) genes in ccRCC tumours.**

Comparison of paired DNA methylation (DNAm) and RNA levels of EMT genes (*IL20RB*, *KRT19*, and *WT1*) in ccRCC tumours in the discovery (IARC ccRCC: 120 tumours; lines and dots in red) and validation cohorts (TCGA-KIRC: 268 tumours) by *BAP1* driver mutation status (wild-type: *BAP1* WT or mutated: *BAP1* MUT). Lines connect matched samples. The boxplots depict the distribution of the data with the central line representing the median (50th percentile). The bounds of the box correspond to the interquartile range (IQR), extending from the 25th percentile (lower quartile) to the 75th percentile (upper quartile). The whiskers show the minimum and maximum values within 1.5 times the IQR from the quartiles, while data points beyond the whiskers are considered outliers. *P* values from Wilcoxon signed-rank. *P* values < 0.05 were considered statistically significant.

