## [Peer Review File · Molecular Systems Biology]

Understanding the biological processes of kidney carcinogenesis: an integrative multi-omics approach

Ricardo CORTEZ CARDOSO PENHA, Alexandra Sexton Oates, Sergey Senkin, Han La Park, Joshua Atkins, Ivana Holcatova, Anna Hornakova, Slavisa Savic, Simona Ognjanovic, Beata Świątkowska, Jolanta Lissowska, David Zaridze, Anush Mukeria, Vladimir Janout, Amelie Chabrier, Vincent Cahais, Cyrille Cuenin, Ghislaine Scelo, Matthieu Foll, Zdenko Herceg, Paul Brennan, Karl Smith-Byrne, Nicolas Alcalá, and James Mckay

Corresponding author(s): James Mckay (mckayj@iarc.who.int)

Review Timeline:

Submission Date:	14th Jun 24
Editorial Decision:	30th Jul 24
Revision Received:	2nd Oct 24
Editorial Decision:	9th Oct 24
Revision Received:	10th Oct 24
Accepted:	21st Oct 24

Editor: Jingyi Hou

Transaction Report:

30th Jul 2024

Manuscript Number: MSB-2024-12479

Title: Disease aetiology and progression shape the inter-patient multi-omics profile of kidney cancer

Author: Ricardo CORTEZ CARDOSO PENHA

Alexandra Sexton Oates

Sergey Senkin

Han La Park

Joshua Atkins

Ivana Holcatova

Anna Hornakova

Slavisa Savic

Simona Ognjanovic

Beata Świątkowska

Jolanta Lissowska

David Zaridze

Anush Mukeria

Vladimir Janout

Amelie Chabrier

Vincent Cahais

Cyrille Cuenin

Ghislaine Scelo

Matthieu Foll

Zdenko Herceg

Paul Brennan

Karl Smith-Byrne

Nicolas Alcalá

James Mckay

Dear Dr. Mckay,

Thank you again for submitting your work to Molecular Systems Biology. We have now finally received the reports from both reviewers and as you will see below, the reviewers think that the presented datasets and analyses seem potentially interesting. They raise however a series of concerns, which we would ask you to convincingly address in a revision.

The reviewers' recommendations are relatively clear, so there is no need to reiterate the points listed below. In particular, Reviewer #1 indicated that the current study does not go deep enough in the interpretability of the observed signals and the roles of the significant features in the disease development and progression. To strengthen the study, we would ask you to improve the data interpretation in the disease context and more clearly highlight the novelty and biological insights of your work.

All other issues raised by the reviewers need to be satisfactorily addressed as well. As you may already know, our editorial policy allows in principle a single round of major revision, and it is therefore essential to provide responses to the reviewers' comments that are as complete as possible.

On a more editorial level, we would ask you to address the following issues:

- Please provide a .docx formatted version of the manuscript text (including legends for main figures, EV figures and tables). Please make sure that the changes are highlighted to be clearly visible.
- Please provide individual production quality figure files as .eps, .tif, .jpg (one file per figure).
- Please provide a .docx formatted letter INCLUDING the reviewers' reports and your detailed point-by-point responses to their comments. As part of the EMBO Press transparent editorial process, the point-by-point response is part of the Review Process File (RPF), which will be published alongside your paper.
- Please note that all corresponding authors are required to supply an ORCID ID for their name upon submission of a revised manuscript.
- We replaced Supplementary Information with Expanded View (EV) Figures and Tables that are collapsible/expandable online (see examples in <http://msb.embopress.org/content/11/6/812>). A maximum of 5 EV Figures can be typeset. EV Figures should

be cited as 'Figure EV1, Figure EV2' etc... in the text and their respective legends should be included in the main text after the legends of regular figures.

Additional Tables/Datasets should be labeled and referred to as Table EV1, Dataset EV1, etc. Legends have to be provided in a separate tab in case of .xls files. Alternatively, the legend can be supplied as a separate text file (README) and zipped together with the Table/Dataset file.

For the figures and tables that you do NOT wish to display as Expanded View figures, they should be bundled together with their legends in a single PDF file called *Appendix*, which should start with a short Table of Content. Each legend should be below the corresponding Figure/Table in the Appendix. Appendix figures and tables should be referred to in the main text as: "Appendix Figure S1, Appendix Figure S2, Appendix Table S1" etc. See detailed instructions regarding expanded view here: <https://www.embopress.org/page/journal/17444292/authorguide#expandedview>.

-Before submitting your revision, primary datasets (and computer code, where appropriate) produced in this study need to be deposited in an appropriate public database (see <http://msb.embopress.org/authorguide> - dataavailability <https://www.embopress.org/page/journal/17444292/authorguide#dataavailability>).

The accession numbers and database should be listed in a formal "Data Availability" section (placed after Materials & Method) that follows the model below (see also <https://www.embopress.org/page/journal/17444292/authorguide#dataavailability>). Please note that the Data Availability Section is restricted to new primary data that are part of this study.

Data availability

-At EMBO Press we ask authors to provide source data for the main figures. Our source data coordinator will contact you to discuss which figure panels we would need source data for and will also provide you with helpful tips on how to upload and organize the files.

- Our journal encourages inclusion of *data citations in the reference list* to directly cite datasets that were re-used and obtained from public databases. Data citations in the article text are distinct from normal bibliographical citations and should directly link to the database records from which the data can be accessed. In the main text, data citations are formatted as follows: "Data ref: Smith et al, 2001". In the Reference list, data citations must be labeled with "[DATASET]". A data reference must provide the database name, accession number/identifiers and a resolvable link to the landing page from which the data can be accessed at the end of the reference. Further instructions are available at .

- We updated our journal's competing interests policy in January 2022 and request authors to consider both actual and perceived competing interests. Please review the policy <https://www.embopress.org/competing-interests> and update your competing interests if necessary.

Please use the heading "Disclosure statement and competing interests".

- All Materials and Methods need to be described in the main text using our 'Structured Methods' format, which is required for all research articles. According to this format, the Methods section includes a Reagents and Tools Table (listing key reagents, experimental models, software and relevant equipment and including their sources and relevant identifiers) followed by a Methods and Protocols section describing the methods using a step-by-step protocol format. The aim is to facilitate adoption of the methodologies across labs. More information on how to adhere to this format as well as a downloadable template (.docx) for the Reagents and Tools Table can be found in our author guidelines: <https://www.embopress.org/page/journal/17444292/authorguide#structuredmethods>.

An example of a Method paper with Structured Methods can be found here: <https://www.embopress.org/doi/10.15252/msb.20178071>.

-Regarding data quantification:

Please ensure to specify the name of the statistical test used to generate error bars and P values, the number (n) of independent experiments (please specify technical or biological replicates) underlying each data point and the test used to calculate p-values in each figure legend. Discussion of statistical methodology can be reported in the materials and methods

section, but figure legends should contain a basic description of n, P and the test applied. Graphs must include a description of the bars and the error bars (s.d., s.e.m.). Please also include scale bars in all microscopy images.

- Please provide a "standfirst text" summarizing the study in one or two sentences (approximately 250 characters, including space), three to four "bullet points" highlighting the main findings and a "synopsis image" (550px width and 400-600 px height, PNG format) to highlight the paper on our homepage.

Here are a couple of examples:

<https://www.embopress.org/doi/10.15252/msb.20199356>

<https://www.embopress.org/doi/10.15252/msb.20209475>

<https://www.embopress.org/doi/10.15252/msb.209495>

When you resubmit your manuscript, please download our CHECKLIST (<https://www.embopress.org/pb-assets/embo-site/EMBO%20Press%20Author%20Checklist-1642513524327.xlsx>) and include the completed form in your submission.

Please note that the Author Checklist will be published alongside the paper as part of the transparent process (<https://www.embopress.org/page/journal/17444292/authorguide#transparentprocess>).

If you feel you can satisfactorily deal with these points and those listed by the referees, you may wish to submit a revised version of your manuscript. Please attach a covering letter giving details of the way in which you have handled each of the points raised by the referees. A revised manuscript will be once again subject to review and you probably understand that we can give you no guarantee at this stage that the eventual outcome will be favorable.

I look forward to receiving the revised manuscript soon.

Kind regards,
Jingyi

Jingyi Hou, PhD
Scientific Editor
Molecular Systems Biology

We realize that it is difficult to revise to a specific deadline. In the interest of protecting the conceptual advance provided by the work, we recommend a revision within 3 months (28th Oct 2024). Please discuss the revision progress ahead of this time with the editor if you require more time to complete the revisions. Use the link below to submit your revision:

IMPORTANT: When you send your revision, we will require the following items:

1. the manuscript text in LaTeX, RTF or MS Word format
2. a letter with a detailed description of the changes made in response to the referees. Please specify clearly the exact places in the text (pages and paragraphs) where each change has been made in response to each specific comment given
3. three to four 'bullet points' highlighting the main findings of your study
4. a short 'blurb' text summarizing in two sentences the study (max. 250 characters)
5. a 'thumbnail image' (550px width and max 400px height, Illustrator, PowerPoint or jpeg format), which can be used as 'visual title' for the synopsis section of your paper.

6. Please include an author contributions statement after the Acknowledgements section (see

<https://www.embopress.org/page/journal/17444292/authorguide>)

7. Please complete the CHECKLIST available at (<https://bit.ly/EMBOPressAuthorChecklist>).

Please note that the Author Checklist will be published alongside the paper as part of the transparent process

(<https://www.embopress.org/page/journal/17444292/authorguide#transparentprocess>).

See also figure legend guidelines: <https://www.embopress.org/page/journal/17444292/authorguide#figureformat>

9. Please note that corresponding authors are required to supply an ORCID ID for their name upon submission of a revised manuscript (EMBO Press signed a joint statement to encourage ORCID adoption).

(<https://www.embopress.org/page/journal/17444292/authorguide#editorialprocess>)

Currently, our records indicate that there is no ORCID associated with your account.

Please click the link below to provide an ORCID:

Link Not Available

11. Include a Reagents and Tools Table as part of the Methods section, which can be downloaded from our author guidelines (<https://www.embopress.org/page/journal/17444292/authorguide#structuredmethods>)

*** PLEASE NOTE *** As part of the EMBO Press transparent editorial process initiative (see our Editorial at <https://dx.doi.org/10.1038/msb.2010.72>), Molecular Systems Biology publishes online a Review Process File with each accepted manuscripts. This file will be published in conjunction with your paper and will include the anonymous referee reports, your point-by-point response and all pertinent correspondence relating to the manuscript. If you do NOT want this File to be published, please inform the editorial office at msb@embo.org within 14 days upon receipt of the present letter.

Reviewer #1:

In this study, the authors generate methylation data for 150 patients with clear cell Renal cell carcinoma (ccRCC) and analyze them jointly with already available WGS and transcriptome data for the same patients. They identify a strong association between the epigenetic clock epiTOC2 and tumor grade, and they report that methylation levels of the xenobiotic metabolism gene GSTP1 link to the tobacco usage. While multi-omics analysis of ccRCC datasets is an interesting and under-explored angle in understanding the disease mechanisms, the authors largely do not go deep enough in the interpretability of the observed signals and the likely role of the significant features in disease development and progression, but rather focus on describing the overall variability in the datasets, which lowers the significance of the study.

Major comments

1. The added value of the multi-omics integration in comparison to an independent metabolome analysis when put in the context of results from the transcriptome and WGS analyses is not clearly shown.
2. The authors do not sufficiently discuss a novelty of their discovery regarding a link between mitotic age and ccRCC.
3. Evidence for several reported observations (BAP1, WT1 genes and other examples) is not clearly and convincingly organized and presented. In addition, relevant mutation and transcriptome results are not always commented in the context of previous studies that analyzed the respective ccRCC data.
4. Pathway analysis - it would be helpful to have the original GSEA pathway names on the Fig before the authors manually reannotated these.
5. TME analysis: It would be good to have a bit clearer analysis of how this was done without going to the original publication and to understand why the used approach was deemed to be better than for instance deconvolution analysis.

Minor comments

1. The authors don't comment on why decided for MOFA as multi-omics integration tool.
2. In my view this abstract sentence lacks a clear message 'Sources of inter-patient variation identified relationships between sex, BAP1 mutations, and epigenetic regulation of epithelial mesenchymal transition genes (e.g., IL20RB, WT1), predicting patient prognosis'. Also, the sentence before this one doesn't send a strong message.
3. I would find the main results section easier to follow if it included a short description on what was done and relevant adjusted p-values for supporting the claims.
4. A short recap on how the explained variance in the previously defined ccRCC clusters was calculated would be helpful (fig 1b).
5. P4 L91 These ccRCC LF signatures were used to infer -> maybe evaluate instead of infer?
6. 164 p 8 different genomic layers -> did you mean different omics layers?
7. P24 L543/544 were represented as sum up by functional weighted -> I didn't understand this (is there a text missing?)
8. P25 549 discover -> discovery
9. Table S8 is listed twice in the suppl table list
10. It is not always clear how strong the evidence is for the features that correlate with the LFs in the validation set if they do not have a predictive value independently.
11. A short comment on the negative values of the variance in Fig 2A would be good to add.
12. A batch correction of the methylation data should be explained in more detail.

Reviewer #2:

In the manuscript titled "Disease Aetiology and Progression Shape the Inter-Patient Multi-Omic Profiling of Kidney Cancer," the authors present a comprehensive study that identifies and characterizes the sources of inter-patient heterogeneity at the multi-omics level. Through integrative analysis of methylome, transcriptome, and somatic mutation profiles using multi-omic factor analysis (MOFA), the authors identified six MOFA factors explaining the major variations in the datasets. These factors are attributed to cellular mitotic age, EMT-related processes, genotoxic effects, etc. The authors validated these factors and their biological/clinical relevance using publicly available external datasets. This study nicely demonstrates the power of utilizing multi-omic data integration to identify novel disease etiologies. The results provide valuable insights into renal cell carcinoma and potentially guide further experimental explorations of the disease mechanism. However, I believe the manuscript can be further improved by addressing a few suggestions/comments below.

1. The authors showed that LF1, LF2, and LF5 have significant associations with patient survival outcomes in both the discovery and validation cohorts. This finding has valuable translational potential and can be strengthened. Can the authors propose a better survival prediction model by combining the three factors in a multiple Cox regression model? Does the model including MOFA factors provide additional predictive power compared to current models based on genomics and clinical parameters, such as mutation markers and tumor stage?
2. As LF2 is potentially related to EMT, does LF2 associate with metastasis, or do the genes/methylation sites contributing to LF2 involve in metastasis?
3. Figure 1A is somewhat confusing. What does the negative variance explained (y-axis) represent? Is it merely to distinguish the Discovery and Validation cohorts, or do the directions of associations differ between the two cohorts?
4. In Figure 3A, it would be better to show the actual data points, not just the regression line, as the regression line does not capture the real pattern of the data and can be misleading.
5. For Figure 4A, if the emphasis is on the negative correlation between methylation levels and gene expression, it would be better to directly use methylation level as the x-axis and gene expression as the y-axis, perhaps using factor 2 value as color annotation. In the legends of both Figures 3A and 4A, the authors wrote "linear regression lines," but the lines do not look like linear regression lines; they seem more like smoothing or local regression lines.
6. In the figure legends (lines 568, 571, 591, 610), "mean" should be "median" or "mean ranks," as the authors were using nonparametric tests (Kruskal-Wallis test and Wilcoxon signed-rank test), which do not test for sample means.

Point-to point answers to reviewers

Reviewer#1:

1.Reviewer#1: “While multi-omics analysis of ccRCC datasets is an interesting and under-explored angle in understanding the disease mechanisms, the authors largely do not go deep enough in the interpretability of the observed signals and the likely role of the significant features in disease development and progression, but rather focus on describing the overall variability in the datasets, which lowers the significance of the study.”

Answer: We would like to thank this reviewer’s overall comment as well as their thorough review of our work. In our revised manuscript, we addressed the reviewer’s concerns by expanding the interpretability of the observed omics signals and how they may act in the kidney cancer development and progression, particularly expanding on these aspects in the updated discussion. This expansion is detailed further in the point-to-point answers below.

2.Reviewer#1:“The added value of the multi-omics integration in comparison to an independent metabolome analysis when put in the context of results from the transcriptome and WGS analyses is not clearly shown.”

Answer: We thank the reviewer for this very pertinent comment. There are two general aspects of added value of integrative multi-omics approach compared to a single-omic approach. First, the integrative multi-omics approach improved the resolution of the signals found within the omics data. Second, the integrative multi-omics approach allowed us to triangulate across the information of the different omics datasets which assisted in our interpretation the observed signals.

In support of the first aspect, we now provide a comparison between the structures identified using PCA analysis of DNA methylation and transcriptome data alone and how they relate to latent factors derived from the integrative multi-omics approach. This comparison demonstrated that only LF1 was resolved with resolution greater than $R^2 > 0.8$ by the single omics approach, with LF2 – LF6 being only partially resolved (Appendix Figure S2).

In response to the second point, since we employed a multi-omics approach from the beginning, we cannot objectively determine whether the insights we identified could have been described using a single-omic layer alone. However, as shown by us and others (Argelaguet et al., 2018: DOI:10.15252/msb.20178124; Cantini et al., 2021, DOI: 10.1038/s41467-020-20430-7), integrating data from multiple omics sources can significantly enhanced the interpretability of the results. For instance, while LF1 was primarily defined by DNA methylation data, its link to the WGS-derived somatic DNA mutation signature (SBS1) and its association with chronological age led us to hypothesize that this structure may be related to biological ageing. Similarly, our interpretations of LF2–LF6 were also strengthened by triangulating insights across multiple omics layers. Therefore, although our assessment may be subjective, there was added value to this integrative approach in understanding the biological mechanisms of kidney cancer.

We have updated the manuscript to make these aspects clearer to the reader. The direct comparison between the single-omic and integrative multi-omics approaches is described in the Appendix Figure S1 and referred to in the text by the following sentence (page 5, lines 108-111): “When compared to the results of PCA analysis derived from either DNA methylation or transcriptome data separately in the discovery cohort, only LF1 was resolved by a single-omic layer analysis (LF1 vs. DNAm_PC1, $R^2=0.96$), while L2-LF6 were partially resolved (Appendix Figure S2)”.

Please, see the Appendix Figure S2 below:

Appendix Figure S2. Comparison of single-omic with integrative multi-omics approaches. Principal component analysis was performed using the same top 5,000 features in each omic layer as those included in the integrative multi-omics approach. The variance across samples explained by each component (PC1-10) was represented in the top left bar plot (DNA methylation data) and in the top right (transcriptome data). Direct comparison between latent factors (Factor1-6) and principal components from DNA methylation (DNAm_PC; bottom left) or transcriptome data (RNA_PC; bottom right). The results were represented as R^2 by exponentiating the pairwise Pearson’s correlation coefficients. Comparisons with Pearson’s correlation $p < 0.05$ were represented.

We have also made the following adjustments in the discussion section (page 13, lines 297-299; page 14, lines 300-310):

“Our study demonstrates the advantages of an integrative multi-omics approach, as emphasized by others (Argelaguet et al., 2018; Cantini et al, 2021). By integrating transcriptome, DNA methylation, and somatic mutation data, we were able to enhance the resolution of molecular structures within patient tumours beyond what single-omics methods could achieve. More importantly, this comprehensive strategy allowed us to extract complementary layers of information, enabling a deeper triangulation of biological meaning in relation to disease mechanisms. We identified key sources of inter-patient variation that not only complemented but also extended previous findings from clustering analyses (Ricketts et al., 2018; Thorsson et al., 2018). For example, LF1, a structure highly correlated with a previously described ccRCC methylation feature (Ricketts et al., 2018), gained added significance in our study when we linked it to the WGS-derived somatic mutation signature (SBS1) and its association with chronological age. This connection led us to explore the hypothesis that LF1 may be related to biological ageing. By integrating such findings with epidemiological and genomic annotations, we gained novel insights into their biological relevance, adding both depth and interpretability to our results.”

3.Reviewer#1:“The authors do not sufficiently discuss a novelty of their discovery regarding a link between mitotic age and ccRCC.”

Answer: To address the reviewer’s comments, we expanded our discussion section regarding the mitotic clocks and ccRCC findings (page 14, lines 311-322; page 15, lines 323-334), as shown below:

“Our approach identified two sources of variance related to biological ageing. The largest source of variance between ccRCC tumours was related to cellular mitotic age, which is a measure of cellular proliferative history. Our study is the first to describe the relationship between different types of mitotic clocks measured across omics layers—namely DNA methylation (epiTOC2), somatic mutations (SBS1), and telomere length in the same tumour sample. The association between LF1 and epiTOC2 was particularly strong. The fact that LF1 levels, which are derived from an unsupervised approach, covaried with epiTOC2 is striking, and it reinforces the hypothesis that the DNA methylation changes at CpG sites of Polycomb target genes might be particularly useful in representing cellular mitotic age. LF1 levels were associated with somatic mutations in PBRM1 and SETD2, chromatin remodeling genes linked to cell senescence (Lee et al, 2016) and proliferation (Cai et al, 2019; Dominguez et al, 2016), which appears consistent with our observation that higher mitotic rates are related to these features. Similarly, LF1 was related to late-stage, high-grade tumours, again consistent with the expected higher mitotic rates (Hakimi et al, 2013; Motzer et al., 2020; Ricketts et al., 2018). Cancer risk factors can affect mitotic rate. For example, higher mitotic rates are observed in histological normal lung epithelial tissue of tobacco smokers compared to never smokers (Young et al, 2018). In this study, the LFs were derived from ccRCC tumours, in that setting the effect of carcinogenesis on the mitotic rate may mask the effects of exposures. Exploring hypothesis that considers how mitotic rates in the

histologically normal kidney are affected by factors like tobacco use, BMI, or acute kidney injury (Kellum et al, 2021) appears warranted. Interestingly, we also distinguished between cellular mitotic age (LF1) and the active proliferation state of the cell (LF5). The later may represent a more aggressive and proliferation-centric tumour phenotype through TP53 mutations affecting directly the cell cycle, while the former could affect indirectly the mitotic rate by regulating the chromatin state through PBRM1 and SETD2 and thereby facilitating access to transcription factors.”

4.Reviewer#1:“Evidence for several reported observations (BAP1, WT1 genes and other examples) is not clearly and convincingly organized and presented. In addition, relevant mutation and transcriptome results are not always commented in the context of previous studies that analyzed the respective ccRCC data.”

Answer: We thank this reviewer for the helpful comment. The first point was also raised by Reviewer#2 (please, see answer 4.Reviewer#2). Following the reviewer#2 suggestion, we have updated Figure 4 to more clearly illustrate the relationship between LF2 and its correlated genes (*IL20RB*, *KRT19*, and *WT1*), showing that the hypomethylation of their regulatory regions were associated with the upregulation of their gene expression. We have also investigated whether *BAP1* driver mutations, which also covaried with LF2, may influence this epigenetic regulation in ccRCC tumours (Figure EV5). Thus, we re-worded the paragraph (page 11, lines 239-252) accordingly:

*“ We performed a differentially methylated region analysis to identify the genomic regions associated with LF2 levels (Table EV10). Among the regions loaded with LF2 levels, the ones showing the highest correlations between DNA methylation and RNA levels mapped to EMT genes (*IL20RB*: $r=-0.84$, $p=4.2 \times 10^{-33}$; *KRT19*: $r=-0.71$, $p=1.2 \times 10^{-19}$; *WT1*: $r=-0.70$, $p=7.1 \times 10^{-19}$) (Table EV10). As LF2 levels increase, the regulatory regions of these genes lose DNA methylation, leading to a corresponding upregulation in their expression (Figure 4A). As *BAP1* somatic mutations were also associated with LF2 levels, we evaluated if this epigenetic effect could be modulated by *BAP1* somatic mutations. Indeed, this epigenetic effect was particularly pronounced for *IL20RB* and *WT1* in the presence of *BAP1* somatic mutations (Figure EV5). As LF2 levels were related to stage IV ccRCC tumours (Figure 4B) and EMT, we evaluated if LF2 and EMT genes were related to metastasis. Higher levels of LF2 were associated with the likelihood of ccRCC patients having distant metastasis ($p=0.007$). This relationship with metastasis was also found with *WT1* ($p=0.017$) and *IL20RB* ($p=7.4 \times 10^{-06}$) expression in ccRCC tumours (Table EV11). ”*

Concerning the second point of this reviewer comment, we added sentences to contextualize our findings with previous transcriptomics and DNA mutation profiling studies, as shown below.

For LF1 and previous DNA methylation clusters (page 9, lines 189-196): The sentence “The above-mentioned LF1 associations provided additional information to the previously described ccRCC methylation clusters in the validation set (TCGA-KIRC, Table EV8)” was replaced by “Interestingly, LF1 levels were partially explained by previously described kidney cancer DNA methylation clusters (Ricketts et al., 2018) ($r^2 = 56\%$, Figure 1B). Both

shared common features, such as the hypermethylation of functional regions in the genome, enrichment for somatic driver mutations (SETD2 and PBRM1), and associations with tumour stage, grade, and overall survival. Nevertheless, multivariable regression analyses showed that LF1 levels could provide additional information in predicting these features when compared to previously described ccRCC methylation clusters in the validation set (TCGA-KIRC, Table EV8)."

For LF5 and previous ccRCC clusters (page 10, lines 220-223): *"When comparing LF5 results (expression of cell cycle genes and TP53 mutations) with previous studies, they resembled those found in the molecular proliferative subgroup of ccRCC patients reported by a previous study (Motzer et al., 2020), both of which associated with poor overall survival of patients."*

For LF2, BAP1, and EMT (page 11, lines 248-254): *"In relation to previous findings, LF2 features partially overlapped with those of two molecular subgroups of ccRCC patients (T-effector/proliferative and stromal/proliferative) identified in a previous study (Motzer et al., 2020), both of which associated with poor survival. This overlap included a higher frequency of BAP1 driver mutations, striking intratumoral adaptive immune responses, and enrichment for gene expression related to the EMT pathway. However, we additionally identified the relationship between the epigenetic regulation of EMT genes and BAP1 somatic mutations in ccRCC."*

5.Reviewer#1: *"Pathway analysis - it would be helpful to have the original GSEA pathway names on the Fig before the authors manually reannotated these."*

Answer: As per the reviewers' suggestion, we included the full GSEA pathway name and identifier in the Table EV4.

6.Reviewer#1: *"TME analysis: It would be good to have a bit clearer analysis of how this was done without going to the original publication and to understand why the used approach was deemed to be better than for instance deconvolution analysis."*

Answer: We appreciate the reviewer's comment. In response we performed a deconvolution analysis and considered how this might differ from our interpretation from the TME/scRNA gene expression signatures analysis. The TME results using the deconvolution tool (R quantiseqr package) in both discovery (IARC ccRCC; N=151) and validation sets (TCGA-KIRC; N=323) are shown below. The results of the deconvolution analysis are largely in line with those from TME gene expression signatures. We included the deconvolution results as Appendix Figure S4 (page 7, lines 144-146): *"As a sensitivity test, we conducted a deconvolution analysis using transcriptome data to infer immune cells signatures and investigate their associations with LF (Appendix Figure S4). The results were largely in line with the those from TME gene expression signatures."*

Appendix Figure S4. Linear regression analyses were used to evaluate the associations between the proportions of immune cells defined using the deconvolution analysis (quantiseq package in R) and latent factors in the discovery (left) and validation (right) sets. The associations were represented as Z-scores (beta estimates divided by the standard errors). Positive associations (Z-score>0). Negative associations (Z-score<0). Blank squares when associations displayed p>0.05. The linear regression model was: Latent factor ~ Immune cell proportion + age at diagnosis + sex.

We included the following sentence in the method section of the revised manuscript to make this point of the TME analysis clearer to the readers (page 24, lines 531-536): “*The inference of TME cells was conducted using pre-defined lists of genes identified in a previous single-cell RNA sequencing study focused on ccRCC tumours (Li et al., 2022). We chose to use these gene lists as they were derived from ccRCC tumours. These gene lists were derived from the Seurat scRNA-seq integration pipeline to integrate ccRCC tumour cells from different patients. Briefly, non-negative factorization analysis was used to cluster ccRCC tumour cells, and gene expression profiles of each cluster were determined. Then, these clusters were assigned to specific cell types or major biological programs within ccRCC based on curated gene sets from external databases.*”

7.Reviewer#1:“The authors don't comment on why decided for MOFA as multi-omics integration tool.”

Answer: Indeed, there are multiple integrative multi-omics tools. Our choice of MOFA was influenced by several aspects the program, particularly that it tolerates missing data values and supports both Gaussian (e.g., array transcriptomics) and non-Gaussian data, aspects important to our study design. Additionally, MOFA has been reported to perform well in clustering samples and its output (latent factors) showed robust associations with genomic annotations in comparison with other integrative tools (Argelaguet et al., 2018:

DOI:10.15252/msb.20178124; Cantini et al., 2021, DOI: 10.1038/s41467-020-20430-7). We included the following sentence in the methods of the revised manuscript to make our choice clearer in the discussion section (page 23, lines 493-498):

“We chose MOFA for our integrative multi-omics analysis because it supports different likelihood models, including both Gaussian (e.g., array transcriptomics) and non-Gaussian models (e.g., presence or absence of driver mutations and DNA mutational signatures), and it also tolerates missing values (Argelaguet et al., 2018). This flexibility allows for a more comprehensive analysis of diverse data types compared to other omics integration tools (Cantini et al., 2021).”

8.Reviewer#1:“In my view this abstract sentence lacks a clear message 'Sources of inter-patient variation identified relationships between sex, BAP1 mutations, and epigenetic regulation of epithelial mesenchymal transition genes (e.g., IL20RB, WT1), predicting patient prognosis'. Also, the sentence before this one doesn't send a strong message.”

Answer: We have re-worded some sentences in the abstract to explicitly describe the key findings of our analyses and send a clear message to the readers in the revised manuscript (page 2, lines 30-44), as mentioned by the reviewer, as shown below:

“Biological mechanisms related to cancer development can leave distinct molecular fingerprints in tumours. By leveraging multi-omics and epidemiological information, we can unveil relationships between carcinogenesis processes that would otherwise remain hidden. Our integrative analysis of DNA methylome, transcriptome, and somatic mutation profiles of ccRCC tumours linked ageing, epithelial-mesenchymal transition (EMT), and xenobiotic metabolism to ccRCC carcinogenesis. Ageing process was represented by associations with cellular mitotic clocks such as epiTOC2, SBS1, telomere length, and PBRM1 and SETD2 mutations, which ticked faster as tumours progressed. We identified a relationship between BAP1 driver mutations and the epigenetic upregulation of EMT genes (IL20RB and WT1), correlating with increased tumour immune infiltration, advanced stage, and poorer patient survival. We also observed an interaction between epigenetic silencing of the xenobiotic metabolism gene GSTP1 and tobacco use, suggesting a link to genotoxic effects and impaired xenobiotic metabolism. Our pan-cancer analysis showed these relationships was also present in other tumour types. Our study enhances the understanding of ccRCC carcinogenesis and its relation to risk factors and progression, with implications for other tumour types.”

9.Reviewer#1:“I would find the main results section easier to follow if it included a short description on what was done and relevant adjusted p-values for supporting the claims.”

Answer: We appreciate the reviewer’s comments. We had presented p-values/Zstats in the figures in order to not overwhelm the reader. However, in recognition to the reviewer’s comments, we have added p-values for discovery and validation cohorts as well as re-worded some sentences to explicitly describe what was done throughout the main text of the revised manuscript to add clarity.

10.Reviewer#1:“A short recap on how the explained variance in the previously defined ccRCC clusters was calculated would be helpful (fig 1b).”

Answer: We have added the following sentence to better explain this point in the legend of Figure 1 (page 29, lines 631-637): *" To estimate the percentage of variance explained in each latent factor (LF1-LF6) explained by the single-omic clusters derived from previous TCGA studies, we applied linear regression models where each latent factor was the outcome and each omic cluster derived from previous TCGA studies was the predictor. This included three DNA methylation clusters and four mRNA clusters (Ricketts et al., 2018), as well as six expression-based immune subtypes (Thorsson et al., 2018). The models provided adjusted R² values, representing the variance explained by each omic cluster in the respective latent factor."*

11.Reviewer#1:“Page 4, line 91. These ccRCC LF signatures were used to infer -> maybe evaluate instead of infer?”

Answer: As suggested by the reviewer, the word ‘infer’ was replaced by ‘evaluate’ in page 5 (line 100) of the revised manuscript.

12.Reviewer#1:“Page 8 , line 164. different genomic layers -> did you mean different omics layers?”

Answer: We apologize for that. The term “genomic layers” was replaced by omics layers throughout the revised manuscript.

13.Reviewer#1:“Page 24, lines 543/544 were represented as sum up by functional weighted -> I didn't understand this (is there a text missing?)”

Answer: We apologise for this lack of clarity. We have updated the following sentences to clarify this point (page 29-30; lines 644; page 30, lines 646-653):

" For the transcriptome layer, pathway analysis was conducted on the top 500 genes correlated with each latent factor using the GSEA database and fgsea R package (v1.27.1). The top 10 enriched biological pathways for each latent factor (FDR < 0.05) were manually categorized into functional groups based on their descriptions in the GSEA database and the canonical functions of the associated genes, including Development, Cell Signaling, Immune System, Chromatin Remodeling, Metabolism, Cell Plasticity, and Cell Cycle. We then summed the normalized enrichment scores, provided by the pathway analysis, by functional category to represent the major biological processes enriched for each latent factor."

14.Reviewer#1:“Page 25, line 549. discover -> discovery”

Answer: Corrected in page 30 (line 659).

15.Reviewer#1:“Table S8 is listed twice in the suppl table list”

Answer: We apologize for this oversight. The supplementary table list was corrected in the revised manuscript.

16.Reviewer#1:“It is not always clear how strong the evidence is for the features that correlate with the LFs in the validation set if they do not have a predictive value independently.”

Answer: We agree that the appropriate resolution of the LF using the signatures in the validation series, which is related to our ability to test for association with risk factors and annotations, is an important aspect of this manuscript. A strategy of validation, calibration and application across three independent datasets would be ideal to accurately estimate the predictive value signatures. In the absence of an ideal third independent dataset, we applied a resampling/bootstrap method approach within the discovery set to estimate the predictive value of our LF signatures (Table EV2). As this approach isn't ideal, we may underestimate the resolution of LF signatures, and thus our ability to test for association with risk factors and annotations in the validation series. Indeed, we noted that several associations were markedly less pronounced than in the validation series. We have point out these limitations in the discussion section (page 18, lines 395-405):

“We used a discovery and validation approach to limit how the play of chance might influence our results and their subsequent interpretation. Nevertheless, differences between the two sets posed limitations to this approach. First, the LF in the validation series were inferred rather than measured directly. The lack of independent sets for training, testing, and validation, ideal for the development of a signature (or biomarker), may have reduced the accuracy of the signature inference and, therefore, the model's ability to capture associations. Similarly, differences in omics data, which include DNA methylation (850k vs. 450k), transcriptome (array vs. RNA-sequencing) and somatic mutation profile (WGS vs. WXS, particularly for DNA mutational signature attribution), might have further contributed to differences in resolution. Indeed, while most features we highlighted here were replicated, some were less pronounced in the validation series, which may be related to this limitation.”

17.Reviewer#1:“A short comment on the negative values of the variance in Fig 2A would be good to add.”

Answer: We appreciate the reviewer's suggestion. This was an oversight of ours. The negative values of the variance had no statistical or biological relevance. Rather the top panel represents the variance explained in the discovery and the bottom in the validation. Figure 1A has been corrected to avoid confusion.

18.Reviewer#1:“A batch correction of the methylation data should be explained in more detail.”

Answer: We added this sentence to make this point clearer for the readers (page 22, lines 484-486):

“The use of the SVA package was performed to remove any potential batch effects from the DNA methylation data, ensuring that any biases introduced by conducting the experiments in two batches did not confound our analysis or influence the results.”

Reviewer#2:

1.Reviewer#2:“In the manuscript titled "Disease Aetiology and Progression Shape the Inter-Patient Multi-Omic Profiling of Kidney Cancer," the authors present a comprehensive study that identifies and characterizes the sources of inter-patient heterogeneity at the multi-omics level. Through integrative analysis of methylome, transcriptome, and somatic mutation profiles using multi-omic factor analysis (MOFA), the authors identified six MOFA factors explaining the major variations in the datasets. These factors are attributed to cellular mitotic age, EMT-related processes, genotoxic effects, etc. The authors validated these factors and their biological/clinical relevance using publicly available external datasets. This study nicely demonstrates the power of utilizing multi-omic data integration to identify novel disease etiologies. The results provide valuable insights into renal cell carcinoma and potentially guide further experimental explorations of the disease mechanism.”

Answer: We appreciate the reviewer's thoughtful comments on our manuscript, particularly on the value and impact of our work.

2.Reviewer#2:“ The authors showed that LF1, LF2, and LF5 have significant associations with patient survival outcomes in both the discovery and validation cohorts. This finding has valuable translational potential and can be strengthened. Can the authors propose a better survival prediction model by combining the three factors in a multiple Cox regression model? Does the model including MOFA factors provide additional predictive power compared to current models based on genomics and clinical parameters, such as mutation markers and tumor stage?”

Answer: We thank the reviewer for this suggestion. We have explored Cox Proportional-Hazards models based on LFs, prognostic factors such as tumour stage, grade, as well as driver somatic mutations (*BAP1*, *SETD2*, and *PBRM1*). There is indeed added value to the LFs in addition to the described factors. We present this in the paragraph (page 12, lines 265-273) in the result section of the revised manuscript:

*“Since LF2 along with LF1 and LF5 levels were associated with patient survival, we generated Cox Proportional-Hazards models to evaluate how they compared to existing models of patient prognosis based on tumour stage, grade and somatic mutations (*BAP1*, *SETD2*, and *PBRM1*) (Table EV13). The combined LF model outperformed the ccRCC driver mutation model (Validation: C-Index=0.71±0.03 vs. 0.63±0.03, Pdiff=0.033) while displaying statistically similar performance to the clinical (C-Index=0.73±0.03, Pdiff=0.560) and integrated models (Validation: C-Index=0.75±0.03, Pdiff=0.215). Given the increment of 2-3% in the performance of the integrated model in relation to the clinical one, our results suggest that these LFs have the potential to complement tumour stage and grade in predicting ccRCC patient prognosis.”*

3.Reviewer#2:“As LF2 is potentially related to EMT, does LF2 associate with metastasis, or do the genes/methylation sites contributing to LF2 involve in metastasis?”

Answer: We thank this reviewer for the comment. As we can see in the figure 4B, the highest levels of LF2 are in the stage IV ccRCC tumours, where the tumours have invasive behaviour (M1). The presence of metastasis in patients was specifically recorded on the TCGA-KIRC cohort, which allowed us to evaluate the relationship between LF2, EMT genes, and metastasis. Indeed, higher levels of LF2 were associated with the likelihood of having distant metastasis. The relationship with metastasis was also found with *WT1* and *IL20RB* expression in ccRCC tumours. We present these new results in Table EV11 and in the main text of the revised manuscript (page 11, lines 243-248):

*“As LF2 levels were related to stage IV ccRCC tumours (Figure 4B) and EMT, we evaluated if LF2 and EMT genes were related to metastasis. Higher levels of LF2 were associated with the likelihood of ccRCC patients having distant metastasis ($p=0.007$). This relationship with metastasis was also found with *WT1* ($p=0.017$) and *IL20RB* ($p=7.4 \times 10^{-06}$) expression in ccRCC tumours (Table EV11).”*

4.Reviewer#2:“Figure 1A is somewhat confusing. What does the negative variance explained (y-axis) represent? Is it merely to distinguish the Discovery and Validation cohorts, or do the directions of associations differ between the two cohorts?”

Answer: We apologise for this lack of clarity in this figure, which was an oversight of ours and also noted by reviewer#1. The negative values of the variance explained for the validation set has no statistical or biological value. They were used for plotting purpose only in order to mirror the values of the discovery set. In order to avoid confusion, we corrected Figure 1A to show positive values for the variance explained for both discovery and validation in the revised manuscript.

5.Reviewer#2:“In Figure 3A, it would be better to show the actual data points, not just the regression line, as the regression line does not capture the real pattern of the data and can be misleading.”

Answer: We updated the figure 3A by including the individual data points in addition to the regression lines for transparency.

6.Reviewer#2:“For Figure 4A, if the emphasis is on the negative correlation between methylation levels and gene expression, it would be better to directly use methylation level as the x-axis and gene expression as the y-axis, perhaps using factor 2 value as color annotation. In the legends of both Figures 3A and 4A, the authors wrote "linear regression lines," but the lines do not look like linear regression lines; they seem more like smoothing or local regression lines.”

Answer: We thank this reviewer for the helpful comments. We updated the figure 4A to highlight the epigenetic regulation of the genes loaded with LF2 that were also mapped to EMT pathways (*IL20RB*, *WT1*, and *KRT19*). We also apologize for the mistakes in the

legends of figures 3A and 4A. We corrected them by replacing “linear regression lines” by “smooth regression lines using the ggplot2 R package.”

7.Reviewer#2:“In the figure legends (lines 568, 571, 591, 610), "mean" should be "median" or "mean ranks," as the authors were using nonparametric tests (Kruskal-Wallis test and Wilcoxon signed-rank test), which do not test for sample means.”

Answer: We apologize for this oversight. We corrected by the word “mean” by “median”.

9th Oct 2024

Manuscript Number: MSB-2024-12479R

Title: Understanding the biological processes of kidney carcinogenesis: an integrative multi-omics approach

Author: Ricardo CORTEZ CARDOSO PENHA

Alexandra Sexton Oates

Sergey Senkin

Han La Park

Joshua Atkins

Ivana Holcatova

Anna Hornakova

Slavisa Savic

Simona Ognjanovic

Beata Świątkowska

Jolanta Lissowska

David Zaridze

Anush Mukeria

Vladimir Janout

Amelie Chabrier

Vincent Cahais

Cyrille Cuenin

Ghislaine Scelo

Matthieu Foll

Zdenko Herceg

Paul Brennan

Karl Smith-Byrne

Nicolas Alcalá

James Mckay

Dear Dr Mckay,

Thank you for sending us your revised manuscript. We have now heard back from the reviewer who was asked to evaluate your revised study. As you will see below, the reviewer is satisfied with the performed revisions and supports publication.

Before we can formally accept the manuscript for publication, we would ask you to address some remaining editorial-level issues listed below.

1. Please reduce the number of keywords to five.
2. Please remove the "Author contribution" section from the manuscript file.
3. The title of conflicts of interest statement should be renamed to "DISCLOSURE AND COMPETING INTERESTS STATEMENT". The "Disclaimer" should be included in "DISCLOSURE AND COMPETING INTERESTS STATEMENT".
4. Section order should be corrected: title page with complete author information, abstract, keywords, introduction, results, discussion, methods, data availability section, acknowledgements, disclosure and competing interests statement, references, main figure legends, tables, expanded figure legends.
5. Please upload the Reagents and Tools table as a separate file choosing the file type "Reagent Table". The template can be found in our author guidelines: <https://www.embopress.org/page/journal/17444292/authorguide#structuredmethods>.
6. Appendix file: title needs to be added to title page (Appendix for Understanding the biological processes of kidney carcinogenesis: an integrative multi-omics approach)
7. Please remove the "Supplemental information" section from the manuscript file.
8. Please note that the Data Availability Section is restricted to new primary data that are produced in the study, all other datasets need to be removed.
9. EV tables

- EV tables should not be uploaded as zip folders, but individual Excel tables.
- Tables EV1-EV5, EV8, EV11, EV13 and EV15 should remain EV tables with updated labels and callouts, but the legends should be only in the main tab, and README sheets should be removed.
- Due to their large size, Tables EV6-EV7, EV9-EV10, EV12, EV14 and EV16 should be renamed to EV Datasets EV1-EVx with corresponding callouts - the legends should remain as separate sheets in each Excel file.

10. Please address the following issues regarding figure legends (main + EV):

- Please note that the exact p values are not provided in the legend of Figure 4A.
- Please indicate the statistical test used for data analysis in the legends of Figures 3a; 4a; 5a; EV 3b.
- Please note that the box plots need to be defined in terms of minima, maxima, centre, bounds of box and whiskers, and percentile in the legends of Figures 3b-c; 4b-c; 5b; EV 5.
- Although 'n' is provided, please describe the nature of entity for 'n' in the legends of Figures 4b; EV 2.
- Please note that the measure of center for the error bars needs to be defined in the legend of Figure EV 2.
- Please note that for heatmap present in Figure 2a, a numbered scale bar is not provided. This needs to be rectified.

When you resubmit your manuscript, please download our CHECKLIST (<https://bit.ly/EMBOPressAuthorChecklist>) and include the completed form in your submission. *Please note* that the Author Checklist will be published alongside the paper as part of the transparent process (<https://www.embopress.org/page/journal/17444292/authorguide#transparentprocess>)

Click on the link below to submit your revised paper.

Kind regards,
Jingyi

Jingyi Hou, PhD
Scientific Editor
Molecular Systems Biology

If you do choose to resubmit, please click on the link below to submit the revision online before 8th Nov 2024.

IMPORTANT: When you send your revision, we will require the following items:

1. the manuscript text in LaTeX, RTF or MS Word format
2. a letter with a detailed description of the changes made in response to the referees. Please specify clearly the exact places in the text (pages and paragraphs) where each change has been made in response to each specific comment given
3. three to four 'bullet points' highlighting the main findings of your study
4. a short 'blurb' text summarizing in two sentences the study (max. 250 characters)
5. a 'thumbnail image' (550px width and max 400px height, Illustrator, PowerPoint or jpeg format), which can be used as 'visual title' for the synopsis section of your paper.
6. Please include an author contributions statement after the Acknowledgements section (see <https://www.embopress.org/page/journal/17444292/authorguide#manuscriptpreparation>)
7. Please complete the CHECKLIST available at (<https://bit.ly/EMBOPressAuthorChecklist>). Please note that the Author Checklist will be published alongside the paper as part of the transparent process (<https://www.embopress.org/page/journal/17444292/authorguide#transparentprocess>).
8. When assembling figures, please refer to our figure preparation guideline in order to ensure proper formatting and readability

in print as well as on screen:

See also figure legend guidelines: <https://www.embopress.org/page/journal/17444292/authorguide#figureformat>

9. Please note that corresponding authors are required to supply an ORCID ID for their name upon submission of a revised manuscript (EMBO Press signed a joint statement to encourage ORCID adoption).

(<https://www.embopress.org/page/journal/17444292/authorguide#editorialprocess>)

Currently, our records indicate that the ORCID for your account is 0000-0002-1787-3874.

Link Not Available

10. Include a Reagents and Tools Table as part of the Methods section, which can be downloaded from our author guidelines (<https://www.embopress.org/page/journal/17444292/authorguide#structuredmethods>)

*** PLEASE NOTE *** As part of the EMBO Press transparent editorial process initiative (see our Editorial at <https://dx.doi.org/10.1038/msb.2010.72> , Molecular Systems Biology will publish online a Review Process File to accompany accepted manuscripts. When preparing your letter of response, please be aware that in the event of acceptance, your cover letter/point-by-point document will be included as part of this File, which will be available to the scientific community. More information about this initiative is available in our Instructions to Authors. If you have any questions about this initiative, please contact the editorial office (msb@embo.org).

Reviewer #2:

The authors have adequately addressed my previous comments. The manuscript in my opinion is suitable for publication.

All editorial and formatting issues were resolved by the authors.

21st Oct 2024

Manuscript number: MSB-2024-12479RR

Title: Understanding the biological processes of kidney carcinogenesis: an integrative multi-omics approach

Dear James,

Thank you again for sending us your revised manuscript. We are now satisfied with the modifications made and I am pleased to inform you that your paper has been accepted for publication.

Kind regards,
Jingyi

Jingyi Hou, PhD
Scientific Editor
Molecular Systems Biology
